# Dissolved greenhouse gases and benthic microbial communities in coastal wetlands of the Chilean coast semiarid region

**Francisco Pozo-Solar[1,2,3], Marcela Cornejo-D´Ottone[4], Roberto Orellana[2,3], Daniela V. Yepsen[5], Nickolas Bassi[6], Julio Salcedo-Castro[7,8], Polette Aguilar-Muñoz[3], Verónica Molina[2,3,9]***

**1** Programa de Doctorado Interdisciplinario en Ciencias Ambientales, Universidad de Playa Ancha, Valparaíso, Chile, **2** Departamento de Ciencias y Geografía, Universidad de Playa Ancha, Valparaíso, Chile, **3** HUB Ambiental UPLA, Universidad de Playa Ancha, Valparaíso, Chile, **4** Escuela de Ciencias del Mar and Instituto Milenio de Oceanografía, Pontificia Universidad Católica de Valparaíso, Valparaíso, Chile, **5** Programa de Doctorado en Ciencias con Mención en Manejo de Recursos Acuáticos Renovables (MaReA), Universidad de Concepción, Barrio Universitario s/n, Universidad de Concepción, Concepción, Chile, **6** Departamento de Geografía, Universidad de Playa Ancha, Avenida Leopoldo Carvallo Valparaíso, Chile, **7** School of Earth and Atmospheric Sciences, Faculty of Science, Queensland University of Technology, Brisbane, Queensland, Australia, **8** Sino-Australian Research Consortium for Coastal Management, School of Science, University of New South Wales, Canberra, Australia, **9** Centro de Investigación Oceanográfica COPAS COASTAL, Universidad de Concepción, Concepción, Chile

* veronica.molina@upla.cl

**Data Availability Statement:** All relevant data are within the paper and its Supporting Information files.

## Abstract

Coastal wetlands are ecosystems associated with intense carbon dioxide ($CO_2$), methane ($CH_4$) and nitrous oxide ($N_2O$) recycling, modulated by salinity and other environmental factors that influence the microbial community involved in greenhouse gases production and consumption. In this study, we evaluated the influence of environmental factors on GHG concentration and benthic microbial community composition in coastal wetlands along the coast of the semiarid region. Wetlands were situated in landscapes along a south-north gradient of higher aridity and lower anthropogenic impact. Our results indicate that wetlands have a latitudinal variability associated with higher organic matter content at the north, especially in summer, and higher nutrient concentration at the south, predominantly in winter. During our sampling, wetlands were characterized by positive $CO_2$ µM and $CH_4$ nM excess, and a shift of $N_2O$ nM excess from negative to positive values from the north to the south. Benthic microbial communities were taxonomically diverse with > 60 phyla, especially in low frequency taxa. Highly abundant bacterial phyla were classified into *Gammaproteobacteria* (*Betaproteobacteria* order), *Alphaproteobacteria* and *Deltaproteobacteria*, including key functional groups such as nitrifying and methanotrophic bacteria. Generalized additive model (GAM) indicated that conductivity accounted for the larger variability of $CH_4$ and $CO_2$, but the predictions of $CH_4$ and $CO_2$ concentration were improved when latitude and pH concentration were included. Nitrate and latitude were the best predictors to account for the changes in the dissolved $N_2O$ distribution. Structural equation modeling (SEM), illustrated how the environment significantly influences functional microbial groups (nitrifiers and methane oxidizers) and their resulting effect on GHG distribution. Our results highlight the

**Funding:** V.M. FONDECYT 1171324, 1211977 Fondo Nacional de Ciencia y Tecnología V.M. and R.O. PAI79170091 Programa de Inserción a la Academia https://www.anid.cl/concursos/ " JS-C. Ministerio del Medio Ambiente (Licitación Pública N° 608897-43-LP81 "Determinación del riesgo de los impactos del Cambio Climático en las costas de Chile" and Centro de Cambio Global UC. FP. Proyecto de Tesis de Postgrado at UPLA. R.O. Grant "Apoyo a la formación de Redes Internacionales para Investigadores en Etapa Inicial" 170600. D.Y. ANID-PFCHA/Doctorado Nacional/2017-21170986. FONDEQUIP EQM160131. The funders had no role in study design, data collection and analysis, decision to publish, or preparation of the manuscript.

**Competing interests:** The authors have declared that no competing interests exist.

combined role of salinity and substrates of key functional microbial groups with metabolisms associated with both carbon and nitrogen, influencing dissolved GHG and their potential exchange in natural and anthropogenically impacted coastal wetlands.

## 1. Introduction

Wetlands are ecosystems that provide critical ecological functions that include organic matter decomposition and nitrogen and carbon recycling, among others. Globally, wetlands act as a source of $CH_4$ and $N_2O$ [1], two of the most relevant greenhouse gasses (GHG). Compared with other aquatic natural sources, wetlands are the ecosystems with the greatest contributions to global $CH_4$ emissions outweighing the contribution of lakes, ponds and other water bodies [2].

Dissolved nutrients and GHG wetland inventories are driven by microbial metabolic processes, showing a temporary (daily, seasonal, monthly) and spatial variability along environmental changes, including solar radiation, temperature, salinity and redox potential, as well as microbial community composition dynamics [3–5]. Microbial processes, such as methanogenesis, are carried out by strict anaerobic archaea that convert acetate, $H_2$, $CO_2$, as well as methylated compounds into $CH_4$, producing up to 90% of total $CH_4$ available on the planet [6]. A diverse and metabolic versatile microbial community composed by methanotrophic bacteria and archaea, oxidizes $CH_4$ [7–9]. Besides oxygen, they can use different electron acceptors, including nitrate [10], sulfate [11], and metals, such as Fe and Mn [12, 13]. On the other hand, $N_2O$ is aerobically produced during the first step of nitrification by ammonia-oxidizing bacteria and archaea or as an intermediate product of denitrification, the anaerobic reduction of $NO_3^-$ to nitrogen gas [14]. The functional microbial communities involved in GHG are often found residing in wetlands sediments where they are subjected to a wide range of environmental fluctuations from natural disturbances to anthropogenic perturbations [3, 15].

The 6,500 km Chilean coastline harbors a great diversity of wetlands, which are widely distributed across different climatic regions influenced by oceanographic processes associated with the Humboldt Current System dynamics [16]. Geophysical processes, such as earthquakes and tsunami events also have a great impact on the coastline, resulting in coastal wetlands with distinctive environmental and biological characteristics, including a high biodiversity and bearing an exceptional endemism [17–19]. Due to this, several of those wetlands have been categorized as Ramsar sites [20]. Wetlands studies are mostly focused in Patagonia fluvial areas and peatlands recently reviewed by León et al. [21]. Based on GHG and microbiological characterization, these ecosystems have been classified as sensitive areas for global warming due to the impact on $CH_4$ emissions mediated by $H_2/CO_2$-based methanogens stimulation [22], highlighting the role of temperature as a driving factor of $CH_4$ emissions globally [23]. Additional studies have provided insights for desert coastal wetlands in Atacama mainly associated with drainage catchment areas and river mouths [24, 25] and hypersaline pools [26], where salinity and substrate availability control methanogenesis/sulfate reduction competition. To our knowledge, there are no studies exploring the relation between GHG distribution and microbial communities in coastal wetlands of the semiarid region of central-northern area, as a transition between the hyper arid and humid areas.

The Semiarid Zone of central-northern Chile (SAZCCh) holds permanent estuary coastal wetlands characterized by the presence of sand bars and water variable salinity, in general, saltier and warmer at north depending on the water sources [27]. This region has a desert-oceanic

Mediterranean bioclimate, with precipitation concentrated in the winter season [28], however, experiencing a mega-drought for the last 30 years [29]. In addition, the wetlands along the Chilean coastline are subjected to variable sources of anthropogenic perturbations, such as, agriculture, mining, tourism, at arid northern area [24] and higher urbanization, litter, industrial and sewage disposal in the central north area [27]. In general, these anthropogenic activities have been shown to considerably affect water quality and quantity [24] and potentially also GHG dissolved concentration in the coastal wetlands semiarid region as reported globally. For instance, the extraction of water and changes of land-use as well as the discharges of agricultural fertilizers have been reported to induce freshwater wetlands salinization [30], triggering wetlands eutrophication, and significantly increasing $CH_4$ and $N_2O$ emissions [3]. The wide range of fluctuating environmental conditions and stressors present in the coastal wetlands belonging to SAZCCh provide a natural platform to explore how the environmental and anthropogenic factors contribute to shaping microbial communities composition and activity as well as the underlying biogeochemical processes related to GHG cycling.

In this study, the distribution of dissolved GHG and the benthic microbial community structure was explored for the first time in a selected group of estuarine coastal wetlands situated in landscapes along a south-north gradient of higher aridity and lower anthropogenic impact in central-northern Chile. Based on these aridity and anthropogenic influence gradient, we addressed the following objectives: 1) To determine physical and chemical variability including dissolved nutrients and GHG, 2) To identify the combination of salinity and other factors accounting for the variability of dissolved GHG, and 3) their role on benthic microbial community composition, including known functional groups. We hypothesize that salinity and latitudinal changes associated with water availability and nutrients enrichments, related with more anthropogenically influenced wetlands (polluted or urban), will increase dissolved GHG inventories by altering the contribution of nitrifiers, methanogens and methanotrophs in the benthic microbial communities. These results will provide for the first time a baseline related to GHG distribution along central-northern Chilean coastal wetlands, highlighting those environmental factors that predominantly affect their dynamic and the responsible key benthic microbial functional groups, serving as guidance for future studies in both natural and impacted coastal wetlands.

## 2. Material and methods

### 2.1 Study area and wetlands of the semiarid zone of central-northern Chile

The group of 24 estuarine type coastal wetlands were selected within the semiarid zone of central-northern Chile (SAZCCh) extending from latitude 29°82´ to 33°77´, considering their environmental surrounding conditions associated with aridity and differential anthropogenic impact (Fig 1, S1 Table). The water source of these wetlands drain mainly from four larger and small catchment areas, Maipo, Aconcagua, Elqui and Limari (Fig 1). Maipo and Aconcagua basins are considered low aridity basins within our latitudinal gradient since they received an annual rainfall average of 20.0 mm and a river discharge of 0.98–46.01 and 14.18–182.16 in Maipo and Aconcagua rivers, respectively (data obtained from the national water monitoring agency, https://snia.mop.gob.cl/BNAConsultas/reportes). In contrast, Elqui and Limarí, as high aridity basins, are characterized by low annual rainfall (average of 0.4 mm) and where their main rivers (Elqui, Grande and Choapa rivers) registered an average annual flow of 3.5 $m^3s^{-1}$ [31]. Moreover, Valparaiso region encompasses more than one hundred glaciers, which represents more than tenfold the glaciers of Coquimbo region [31].

**2.1a Wetlands from high aridity basins (HAB).** The wetlands sampled from HAB are located within the Coquimbo geopolitical region and are subjected to a variable anthropogenic

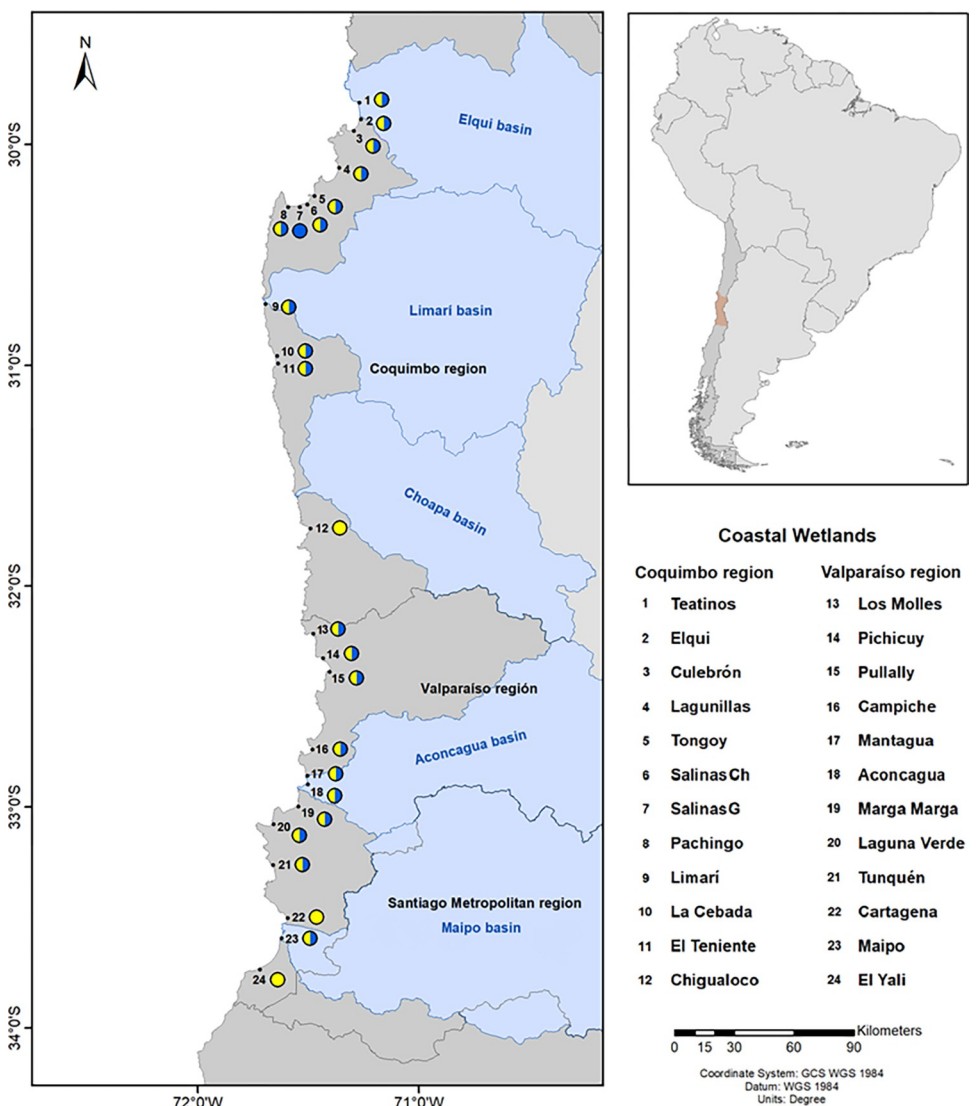

**Fig 1. Study area map showing the coastal wetlands sampled through latitudinal gradient in winter and summer (blue/yellow circles).** Cartagena and Yali wetlands were only sampled in summer (yellow circles) and Salinas G wetland was only sampled in winter (blue circles). Map was created using cartography Shapefiles publically available, i.e., "Basin Layer" from the Water Resources Directorate (Dirección General de Aguas, DGA) and "Administrative layer" the Subsecretary of Regional Development (Subsecretaría de Desarrollo Regional y Administrativo (SUBDERE). The wetland geographical position was mapped using QGIS, open code version 2.18 version software (https://qgis.org/downloads/).

perturbation. For instance, Punta Teatinos, the mouth of Elqui River and El Culebrón associ-ated with the Coquimbo bay (Fig 1, S1 Table) are impacted by the fast urbanization and expan-sion of Coquimbo and La Serena cities (S1 Fig). Whereas, another set of wetlands located further south, such as, Lagunillas, Estero Tongoy, Salinas Chica, Salinas Grande and Pachingo, are characterized by high biological diversity, some listed in RAMSAR category (see links to Ramsar sites in S1 Table). In general, the vegetation of these coastal wetlands is mainly domi-nated by *Sarcocornia fruticosa* (Salinas) and *Typha angustifolia* (Tifales) [32].

**2.1b Wetlands from low aridity basins (LAB).** The wetlands sampled from LAB are located within the Valparaíso geopolitical region (Fig 1, S1 Table). Wetlands belonging to LAB are characterized by higher anthropogenic impact, including agricultural, industrial, sewage

and different types of contamination in the main rivers Aconcagua and Maipo, and other estuaries such as Los Molles, Campiche, Laguna Verde and Cartagena [33]. Moreover, this area concentrates a larger population and coastline causing a great impact on the coastal wetlands [17]. Only Tunquén and El Yali were classified as class II, attractive sites with high landscape value and natural reserve, also listed in RAMSAR [20] (S2 Fig).

## 2.2 Field sampling campaigns in the semiarid zone of central-northern Chile

Winter field campaigns in the Valparaiso Region (southern basins) were conducted on July 28, 29 and 30, and in the Coquimbo Region (northern basins) on August 25, 26 and 27, 2017. In summer, samples were collected in the Valparaiso Region on December 27, 28 and 29, 2017 and in the Coquimbo Region on January 3, 4 and 5, 2018. A summary of the wetland geographic positions, sampling dates and all the data available for our study for winter and summer seasons is included in the S1 Table.

All samples were taken from the estuary mouth in a single site at a low downstream flow area. All estuaries sampled are freely accessible through their mouths that adjoin public beaches. To visit natural protected areas with protection such as Yali permits were given for the Universidad de Playa Ancha by the Corporación Natural Forestal (CONAF) between the years 2014 and 2019 considering the policy of the national regulation of natural protected zones (Sistema Nacional de Áreas Silvestres Protegidas del Estado, SNASPE). The sampling was carried out wearing waders, using a standard procedure in the different wetlands, consisting in standing against the downstream flow to avoid sediment resuspension and sample contamination at a comparable depth of approximately 40–60 cm, without the influence of seawater intrusion or tides. Water surface temperature, pH, dissolved oxygen (DO) and conductivity were determined *in-situ* using a multiparameter (probe YSI and ProDSS model) at a single point. Water samples for nutrient determination, i.e., nitrate ($NO_3^-$), nitrite ($NO_2^-$), phosphate ($PO_4^{3-}$) and silicic acid ($SiO_4^{4-}$), were collected in triplicate directly from the surface using a syringe and immediately filtered (0.45 μm GFF) in "Nalgene" bottles (60 mL). These samples were transported at 4˚C during field campaigns and then frozen until analyzed at the laboratory. To determine GHG ($CO_2$, $N_2O$ and $CH_4$) concentrations, water samples were collected in triplicate using 20 mL gas tight vials, avoiding bubbles, immediately fixed with 50 μL mercury chloride (saturated) and hermetically sealed. The vials were stored at room temperature in the dark until performing the analyses at the laboratory. Samples for GHG analyses were collected in summer in all regions, and in winter only in the Valparaiso Region. Fine grain wet-sediment samples (occupying an approximate volume of 300–500 μL) were collected at a single point in cryotubes (1.5 mL) filled with RNA-later (Ambion, Life Technologies, USA), a buffer used to preserve both RNA and DNA (Gray et al., 2013) [34]. These samples were transported at 4˚C to the laboratory and stored at -80˚C until subsequent DNA extraction. Sediment sampled in the winter campaign from Salinas G, El Teniente and Chigualoco (Coquimbo Region) and Pichicuy, Laguna Verde Cartagena, Tunquén and El Yali (Valparaiso Region) and sediment sampled in the summer campaign from El Teniente and Chigualoco (Coquimbo Region) and Pichicuy, Marga Marga, Laguna Verde and Tunquén, were excluded from nucleic acid extraction because they contained high levels of very coarse grain, which could bias further analysis.

## 2.3 Chemical analyses determined in the water and sediments of the wetland sampled

Nutrient concentration in the water was determined using spectrophotometric methods. $NO_2^-$ concentration was analyzed as indicated by Strickland and Parsons (1972), whereas $NO_3^-$, $PO_4^{3-}$ and $SiO_4^{4-}$ were measured using a nutrient autoanalyzer [35].

Dissolved GHG ($CO_2$, $CH_4$ and $N_2O$) concentrations were determined by gas chromatography using the headspace technique [36], in a gas chromatograph (Greenhouse GC-2014, Shimadzu). The chromatograph was equipped with an electron capture detector (ECD) for $N_2O$ and a flame ionization detector (FID) for $CO_2$ and $CH_4$ determinations, attached to a methanizer for the conversion of $CO_2$ to $CH_4$.

GHG contribution ($\Delta[A]$) was estimated based on discrete GHG determination, following Sarmiento and Gruber (2004) [37]:

$$\Delta[A] = [A]_w - [A]_{equilibrium}$$

were, $[A]_w$ is the gas concentration in the water and $[A]_{equilibrium}$ is the equilibrium with the atmosphere gas concentration. The last one is calculated with the following formula:

$$[A]_{equilibrium} = S_A \; x \; p^A$$

were, $S_A$ is the solubility of the gas in the water A and $p^A$ is the partial pressure of the gas A in the atmosphere. This calculation allowed us to classify each wetland as a potential source or sink for atmospheric GHG. Therefore, when $\Delta[A] > 0$, the wetland is a source of GHG, and if $\Delta[A] < 0$, the wetland is a sink of GHG.

Surface sediment samples were collected in sealable polyethylene bags at a single point in each wetland and transported at 4°C to the laboratory and stored at -20°C to the laboratory until analysis within a month. Organic matter was determined by the loss on ignition (LOI) method in a Muffle furnace. For granulometry determination, the samples were dried at 40°C and then analyzed using the standard method for mesh analysis to separate different sizes of sand, gravel, and silt.

## 2.4 Benthic microbial community composition characterization in the wetlands using 16S rRNA gene

Nucleic acid extraction and sequencing of 16S rRNA gene were used for benthic microbial community characterization. After RNA-*later* removal, 250 mg of wet sediment were weighted and then DNA was extracted using the DNaeasy Power Soil DNA Isolation kit (MoBio Laboratories), following the manufacturer´s instructions. DNA extracts were quantified with the dsDNA BR (Broad Range) Qubit 2.0 fluorometer (Thermo Fisher Scientific) and the DNA quality (260/290) was measured by spectrophotometry using Cytation Take3 nanodrop (Bio-Tek). To verify 16S rDNA amplification, a PCR was carried out using the following primers: 27F (5´-AGRGTTYGATYMTGGCTCAG-3´) and 519R (5´-GWATTACCGCGGCKGCTG-3´). PCR conditions were the following: 95°C for 5 min followed by 35 cycles of 95°C for 45 s, 56°C for 45 s, and 72°C for 45 s, and a final elongation step of 5 min at 72°C using GoTaq® DNA Polymerase (Promega Corporation). PCR products were checked using electrophoresis for 45 min at 70 volts in an agarose gel (1,5%). DNA extracts were sequenced using Illumina Miseq sequencing platform at Mr. DNA laboratory (Texas, USA), using in-house primers 515F (GTGYCAGCMGCCGCGGTAA) [38] and 806R (GGACTACNVGGGTWTCTAAT) [39] for the V4 region of the 16S rDNA gene. Sequences were deposited in the European Nucleotide Archive (ENA) under the project accession ID PRJEB44346 and the primary accession samples (ERR5725941- ERR5725957).

Sequencing data -processing analyses- were carried out using QIIME2 (v.2019.7) for sequences curation and taxonomic classification into ASVs (Amplicon Sequence Variant) [40]. First, sequences were imported to QIIME2 using Casava. The demultiplexed, poor quality and short sequences were extracted (S2 Table), leaving c.a. 210 bp length sequences, using *qiime dada2 denoised-paired* plugin [41]. The taxonomic classification was performed using

the SILVA132 database (release 132–99% OTUs, 515–806 region) with the *feature-classifier classify-consensus-vsearch* plugin [42]. Then, readings on abundance of known functional groups taxa associated with ammonia and methane recycling mainly related with oxidation metabolic pathways, known to produce $N_2O$ or consume $CH_4$ respectively, were selected for their subsequent modeling with the environmental factors.

Alpha diversity analysis of the microbial community was estimated using ASVs (diversity, richness, and evenness). Bray-Curtis similarity analyses were performed at different taxonomy levels (phyla and ASVs) using squared-root transformed data. Principal Coordinates Analysis (PCO) was generated to assess changes in the microbial community´s structure and its relationship with the environmental factors using Primer6 software [43]. Similarity Percentage (SIMPER) was estimated to identify the phyla and ASVs contribution to similarity/dissimilarity between seasons (winter and summer), saline category (hyposaline and mesosaline wetlands) and arid category.

## 2.5 Statistical analysis to identify environmental factors influence on GHG and benthic microbial community

To determine spatial and seasonal changes of the environmental factors, nutrients and OM a two-way ANOVA was used. Spearman multiple correlation analyses were carried out between conductivity and other environmental factors, with the GHG and functional groups, since most of the variables did not present a normal distribution (Shapiro-Wilk normality test, $p < 0.05$). A Non-metric multidimensional scaling (NMDS) was conducted to determine the influence of the sampled seasons (winter versus summer) as a factor, based on the stress level of NMDS fit, which shows the relationship between ordination distance and observed dissimilarity. It provides a flexible framework for modeling the relationship between individual predictors and dependent variables. Then, GAM was used to determine the contribution of conductivity with other environmental factors on GHG and functional groups variability. The "mgcv" package [44] and the R statistical software [45] were utilized.

GHG response was evaluated using the following linear predictor:

$$GHG_i = \mu + s(C) + s(lon) + s(lat) + s(t) + s(pH) + s(d\_oxy) + s(nitrate) + s(p) + s(s) + s(nitrite) \sim$$

$$Negative\ Binomial$$

were $GHG_i$ is gas concentration; $i$ $\mu$ is the intercept; $s()$ is the spline softener; $C$ is the conductivity effect on gas; $i$, $lat$ is the latitude effect on gas; $i$, $lon$ is the longitude effect on gas; $i$, $t$ is the water temperature effects on gas; $i$, $pH$ is the pH effect on gas; $i$, $d\_oxy$ is the dissolved oxygen effects on gas; $i$, $nitrate$ is the nitrate effect on gas; $i$, $p$ is the phosphate effect on gas; $i$, $s$ is the silicic acid effect on gas; $i$, $nitrite$ is the nitrite effect on gas $i$. For the dependent variable, a negative binomial error distribution with a link type "log" was used. GHG models were built on N = 32.

The functional group response was evaluated using the following linear predictor:

$$taxa\_i = \mu + s(C) + s(lon) + s(lat) + s(t) + s(pH) + s(d\_oxy) + s(nitrate) + s(p) + s(s) + s(nitrite) \sim$$

$$Negative\ Binomial$$

were $taxa_i$ is taxa $i$, $\mu$ the intercept; $s()$, the spline softener; $C$ is the conductivity effect on taxa; $i$, $lat$ is the latitude effect on taxa; $i$, $lon$ is the longitude effect on taxa; $i$, $t$ is the water temperature effects on taxa; $i$, $pH$ is the pH effect on taxa; $i$, $d\_oxy$ is the dissolved oxygen effects on taxa; $i$, $nitrate$ is the nitrate effect on taxa; $i$, $p$ is the phosphate effect on taxa; $i$, $s$ is the silicic

acid effect on taxa; *i*, *nitrite* is the nitrite effect on taxa *i*. For the dependent variable, a negative binomial error distribution with a link type "log" was used. Taxa models were built on N = 34, except the models that included nitrite, since they had N = 24.

Structural equation modeling (SEM) was used to determine multivariate causal relationships by considering key factors previously identified by GAM as significant to explain either the variability of GHG and specific functional groups. Those factors include latitude, pH, conductivity, nitrate and organic matter. The resulting latent variables were "environmental variables, ENV", "functional groups, FG" and "GHG", and the structural relationship between them was explored considering as a theoretical causal model that ENV influences FG community structure resulting in changes in GHG distribution. SEM model was run by means of lavaan project (ugent be) in R studio, using diagonally weighted least squares (DWLS) estimation to fit the weight covariance matrix using only the diagonal of this matrix and the NLMINB optimization method. Chi-square (p>0.05) test and root mean square error of approximation (RMSEA), comparative fit index (CFI) and Tucker-Lewis Index (TLI) were included to evaluate the model fit.

## 3. Results

### 3.1 Coastal wetlands physical and chemical conditions in the water and sediments of the latitudinal gradient of the semiarid zone of central-northern Chile

The physical and chemical conditions in the water and sediments varied depending on the wetland geographic localization and the season sampled. The average temperature ranged between 11 and 30.5°C, reaching significantly higher values (p<0.001) in the high aridity basins (HAB) compared with the low aridity basins (LAB) and temperature was significantly greater in summer (p<0.01) unlike winter (Fig 2A; Table 1). Other variables, such as pH ranges (7.18–10.38 pH units) registered significantly higher values (p<0.001) in summer compared to winter (Fig 2B; Table 1). Conductivity was the environmental factor that showed the highest variability, ranging from 0.550 to 42.528 and from 1.162 to 63.519 mS/cm in winter and summer, respectively, with significantly higher values (p<0.05) in the HAB versus LAB (Fig 2C, Table 1). In addition, DO ranged from 4.27 to 19.34 mg/L in winter and from 6.9 to 17.18 mg/L in summer (Fig 2D). Differences among dissolved nutrients across seasons were not observed, however, nitrate and silicic acid showed significantly greater values in LAB compared to HAB (Fig 2E and 2H, Table 1). Phosphate and nitrite registered low variability (Fig 2F and 2G) and phosphate did not show significant spatial and seasonal changes (Table 1), but reached maximum concentration in polluted wetlands, in winter and summer. Indeed, waters from Maipo, Laguna Verde, Aconcagua, Marga Marga and Elqui, all estuaries located close to urban or industrial areas, registered higher concentrations of phosphate (20–35 μM), nitrate (100–500 μM) and/or nitrite (10–70 μM) than the other wetlands (S1 Table).

The wetland sediment granulometry was associated with different grain sizes ranging from silt to very coarse sand types (S3 Fig). The high level of silt and very fine sand sediments were characteristic of wetlands from HAB, and only a few from LAB, mainly in summer (S3B Fig). Besides, HAB wetlands showed significantly higher organic matter content (> 4 g), in addition organic matter content was significantly greater in summer compared to winter (S3 Fig and Table 1).

Dissolved GHG concentration in the wetlands revealed a broad range: $CH_4$, 4.4–1206 nM, $CO_2$, 0.6–507.2 μM and $N_2O$, 3.7–83.3 nM (90–1,941 ppb) (S1 Table). $CO_2$ presented significantly higher concentrations in winter than in summer (One-Way ANOVA p<0.001) in LAB wetlands. In general, most wetlands are $CO_2$ sources, except for six wetlands sampled in

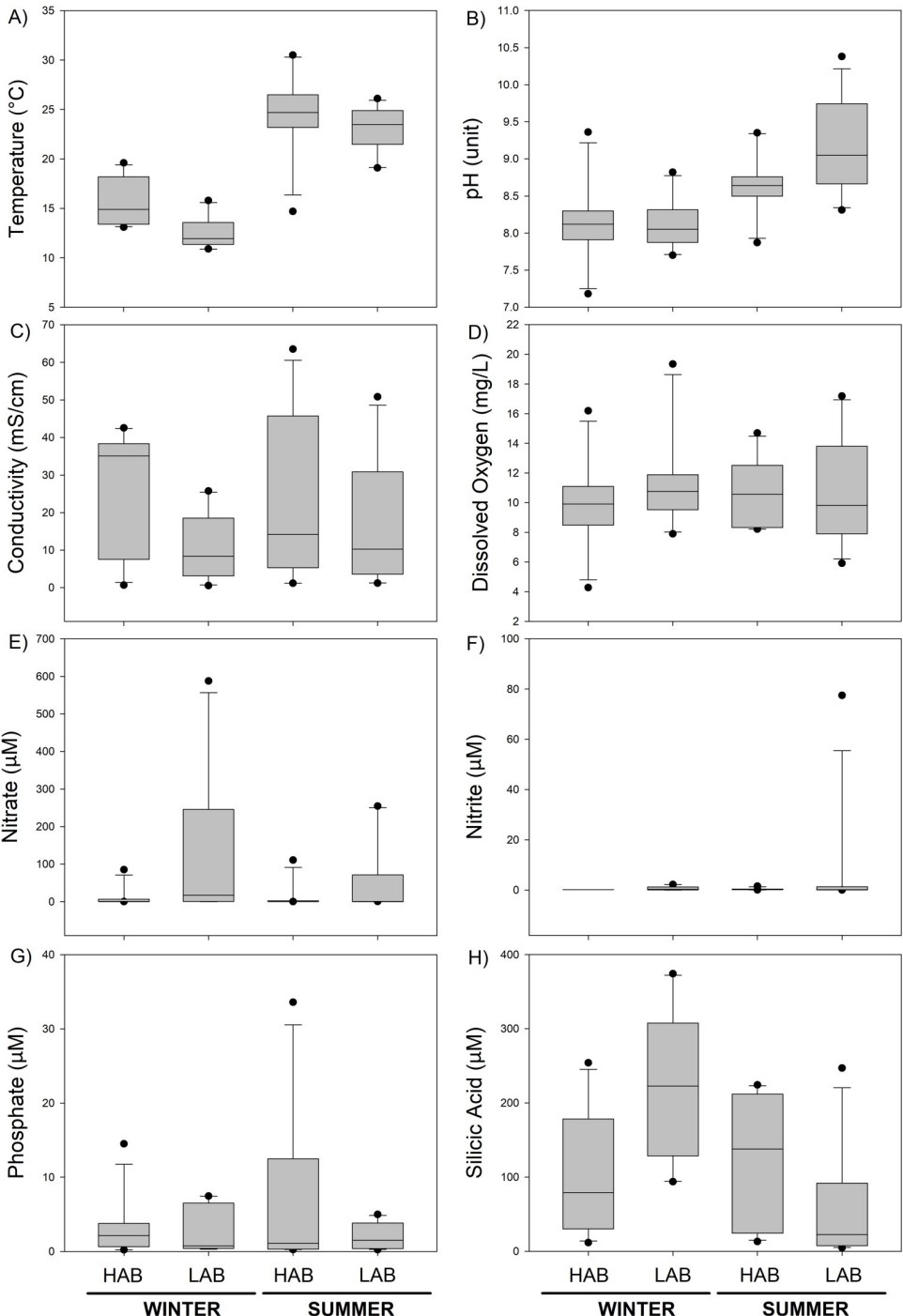

**Fig 2. Physical and chemical conditions and dissolved nutrients determined in high versus low aridity basins wetlands water during summer and winter.** Outliers are indicated as black dots. The detailed table containing data for each wetland can be found in S2 Table.

summer, i.e., Chigualoco, Los Molles, Pichicuy, Campiche and Yali, (Fig 3A). All wetlands registered an excess of $CH_4$, except Limarí, which showed dissolved concentration in equilibrium with the atmosphere (Fig 3B). $N_2O$ was the only GHG showing a marked gas excess shifts

**Table 1. Two-way ANOVA comparing physical and chemical conditions, dissolved nutrients and organic matter determined in high versus low aridity basins wetlands water during summer and winter.** Significant values (p <0.05) are highlighted in grey.

| Variable | Season | HAB vs LAB | Shapiro-Residuals | Levene |
|---|---|---|---|---|
| Temperature | 4.86 E-03 | 1.38 E-14 | 9.35 E-03 | 3.98 E-01 |
| pH | 5. 08 E-06 | 1.15 E-01 | 1.51 E-01 | 6.28 E-01 |
| Conductivity | 4.38 E-01 | 4.51 E-02 | 6.71 E-02 | 1.44 E-01 |
| Dissolved Oxygen | 8.34 E-01 | 4.25E-01 | 3.97 E-02 | 8.12 E-01 |
| Nitrate | 2.37 E-01 | 2.25 E-02 | 3.58 E-07 | 4.34 E-02 |
| Phosphate | 7.28 E-01 | 8.31 E-02 | 8.24 E-09 | 3.75 E-01 |
| Silicic Acid | 1.60 E-01 | 1.42 E-02 | 5.46 E-02 | 5.86 E-02 |
| Organic Matter | 4.717 E-03 | 4.10 E-06 | 4.55 E-02 | 3.06 E-03 |

across latitude from positive to negative values, suggesting that LAB wetlands act as a potential $N_2O$ source, whereas those LAB wetlands, except for the Elqui, act as an $N_2O$ sink (Fig 3C). Interestingly, Laguna Verde (in summer) and Maipo (in winter), two of the wetlands most severely impacted by pollution, registered the highest magnitude of $N_2O$ excess (Fig 3C).

## 3.2 Relationship between environmental factors, nutrients and their modulating role on GHG distribution

Spearman multiple correlation analysis indicates that conductivity presented a high number of significant correlations with GHG concentration, including silicic acid (r = -0.502, p < 0.001), nitrate (r = -0.488, p < 0.001), $CO_2$ (r = -0.624, p < 0.001), $CH_4$ (r = -0.400, p < 0.05) and $N_2O$ (r = -0.495, p < 0.01) (S3 Table). Water temperature was significantly correlated with $N_2O$ (r = -0.632, p < 0.001) and pH (r = 0.59, p < 0.001) and pH was correlated with $CO_2$ (r = -0.704, p < 0.001) and dissolved oxygen (r = 0.424, p < 0.01) (S3 Table). In addition, the sediment organic matter (OM) was significantly correlated with temperature (r = 0.463; p <0.01), $N_2O$ (r = -0.686; p < 0.001) and very fine sand (r = 0.439; p < 0.01) and very coarse sand (r = -0.396; p <0.05) (S3 Table).

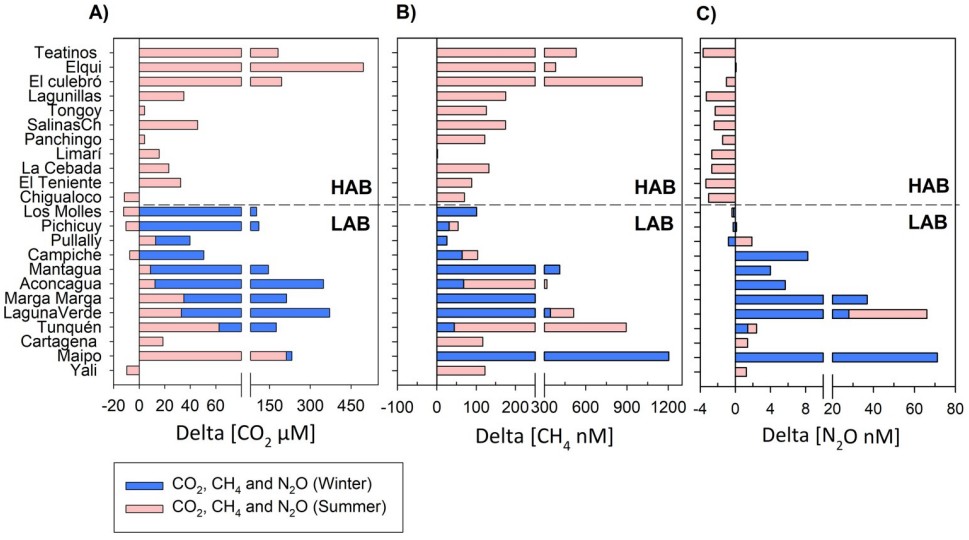

**Fig 3.** $CO_2$ (A), $CH_4$ (B) and $N_2O$ (C) greenhouse gases "excess" that were estimated from original concentration in the water in winter (pink bars, only for LAB wetlands) and summer (blue bars, for HAB wetlands).

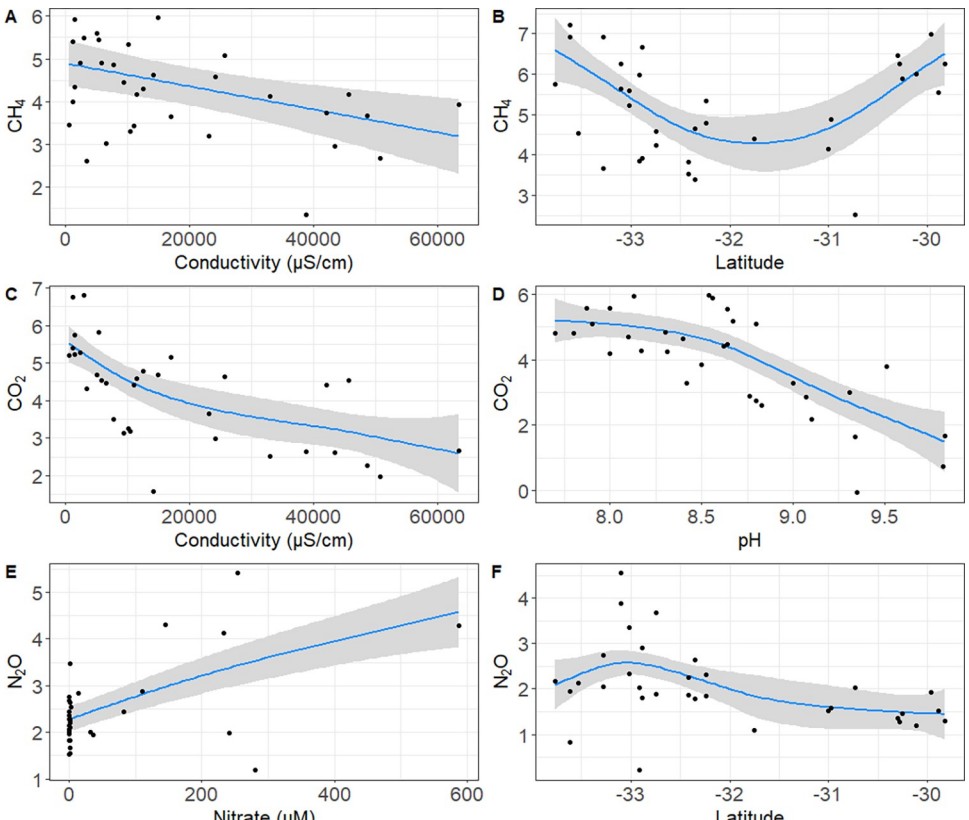

**Fig 4.** GAM models showing the relationships between the greenhouse gas distribution and different environmental factors: $CH_4$ with conductivity (A), Latitude (B), $CO_2$ with conductivity (C) and pH (D) and $N_2O$ with nitrate (E), and latitude (F). Axes "Y" correspond to the effect of the different predictor variables.

Generalized additive models (GAM) indicate that among all the variables analyzed, conductivity and latitude were significant predictors of $CH_4$ distribution in the wetlands, accounting for 60.7% and 415.6 of deviance and AIC, respectively (Fig 4A and 4B; Table 2). In the case of $CO_2$, conductivity and pH were explanatory variables with 81.4% and 330.7 of explained deviance and AIC, respectively (Fig 4C and 4D; Table 2). Regarding $N_2O$, conductivity was not a good predictor to account for changes in its distribution across wetlands. The two most significant predictors to account for changes in $N_2O$ concentration were latitude and nitrate, representing 89.4% of the deviance and 191.5 of AIC (Fig 4E and 4F; Table 2).

### 3.3 Microbial community structure and composition characteristics in the coastal wetlands

In general, alpha diversity estimators indicate that there was a higher variability in winter than in summer, i.e., richness (201–675 versus 416–593 ASV), diversity (3.784–5.237 versus 4.236–5.193 H´) and evenness (0.62–0.80 versus 0.66–0.81), (S4 Table). A discrete set of wetlands, including Limarí, La Cebada, Mantagua, Aconcagua and Maipo, presented contrasting seasonal changes, characterized by the highest richness (> 550 ASVs) in winter than in summer (S4 Table). Principal Coordinate Analysis (PCoA) allowed us to visualize the variability of microbial community structure at the ASV level, for which PC1 and PC2 accounted for c.a. 32.7% of the total variability (S4 Fig). The overlaid correlated environmental variables visualized as vectors highlighted that the microbiome structure was mostly influenced by conductivity (Spearman correlation r = 0.644

with PC1), categorized according to their salinity i.e., mesosaline versus hyposaline ($> 20$ mS/cm vs $< 20$ mS/cm). Nitrate was also significantly correlated with PC1 (r = -0.312), likewise, temperature (r = 0.619), organic matter (r = 0.522) and silt (r = -0.370) correlated well with PC2.

The microbial composition analysis of wetland sediments showed that bacteria (2,821,908 total reads) predominate over archaea (70,397 total reads), representing 97.53% and 2.47%, respectively. The bacterial community was associated with 64 phyla, among which 49 of them contributed less than 1% of the total retrieved sequences (S5 Fig). The most abundant bacterial phylum was *Proteobacteria*, showing a winter versus summer variability; in the summer, these changes include a slight decrease in the average relative abundance of *Gammaproteobacteria* and *Alphaproteobacteria* from 16.26 to 13.04% and from 10.79 to 9.33%, respectively. In contrast, *Deltaproteobacteria* increased smoothly in summer compared with winter (8.76 versus. 11.41%) (S5 Table). Other abundant phyla in the wetlands were *Bacteroidetes* and *Cyanobacteria*, but only the latter showed higher abundance in summer (11.17%) compared with winter (7.82%). When considering wetlands salinity, *Gammaproteobacteria* class was the most abundant taxa contributing to 15.55% in hyposaline wetlands and 13.82% in mesosaline wetlands (S5 Table). On the other hand, archaeal community was composed of 9 phyla mainly affiliated to *Thaumarchaeota*, *Nanoarchaeota* and *Euryarchaeota* (S6 Fig). SIMPER analysis indicated that in winter *Thaumarchaeota* and *Nanoarchaeota* account for 68.1 and 18.14% of the total archaeal sequences, while in summer their contribution was 24.76 and 66.09%, respectively. SIMPER analysis using salinity predicted slightly higher percentages of contribution to the archaeal sequences in hyposaline compared with mesosaline wetlands for *Thaumarchaeota* (49.37 vs. 45.73%) and *Euryarchaeota* (12.35 vs. 11.76%). In contrast, the opposite was found for *Nanoarchaeota*, which showed an increase from 37.65% to 41.9% (S6 Fig and S5 Table).

## 3.4 Relationship between environmental factors, nutrients, and factors reshaping microbial functional groups

The microbial communities associated with wetlands were characterized by many functional groups, including archaea and bacteria ammonia oxidizers, methanogens and methanotrophs, which significantly correlates with different environmental factors sampled (S6 Table). *Nitrosomonas* sp. (r = 0.67) and other genera affiliated with nitrifying bacteria including AOB and *Nitrospira* sp. (r≥0.55) presented a significantly high correlation with $N_2O$ (S6 Table). Ammonia and methane oxidizing bacteria belonging to *Betaproteobacteria* (*Nitrosomonas* sp. and *Methylotenera* sp., respectively) and the nitrite oxidizing bacteria (*Nitrospira* sp.), were the only genera significantly correlated with conductivity and frequent enough to allow modeling with environmental factors including conductivity (S1 Table, Table 3). Moreover, summer and winter data were combined considering that non-metric multidimensional scaling analysis indicated that season (summer versus winter) was not an influencing factor of the structure of the mentioned functional groups (S7A and S7B Fig). GAM results generated with a selected microbial functional groups considering their representation ≥28 wetlands (S1 Table) indicated that potential factors accounting for *Betaproteobacteria* order distributions were conductivity and OM and nitrate (Fig 5A and 5B), based on the deviance and AIC obtained (Table 3).

In addition to that, SEM model analysis was performed using conductivity, pH, latitude, season, nitrate and organic matter to determine potential multivariate causal relationship on the variability of a selected microbial communities known for their contribution on ammonia and methane oxidation named as functional groups (FG) and then identify the FG influence on GHG using a path diagram presented in Fig 6. The result showed that latent variables (ENV, Environment; FG, Functional Group; GHG, Greenhouse gases) were differentially and significantly affected by the observed factors (Fig 6). A relationship between the latent variables

**Table 2. GAM models using environmental factors with nutrients as predictor variables and greenhouse gases as response variables.**

| GHG | Models | AIC | ED | p-value |
|---|---|---|---|---|
| $CH_4$ | ~ s(C) +s(Lat) | 415.6 | 60.7 | C = 1.65 E-03; Lat = 4.53 E-06 |
| | ~ s(C) | 430.9 | 34.2 | C = 4.33 E-04 |
| $CO_2$ | ~ s(C) +s(pH) | 330.7 | 81.4 | C = 2 E-16; pH = 2 E-16 |
| | ~ s(C) | 364.6 | 31.4 | C = 1.06 E-05 |
| $N_2O$ | ~ s(Nitra) +s(Lat) | 191.5 | 89.4 | Nitra <2 E-16; Lat = 5.66 E-05 |
| | ~ s(C) +s(Nitra) | 213.2 | 74.7 | C = 1.05 E01; Nitra = 2 E-016 |
| | ~ s(Nitra) | 213.3 | 73.2 | Nitra <2 E-16 |
| | ~ s(Nitri) | 225.4 | 62.5 | Nitri <2E-16 |
| | ~ s(C) +s(Nitri) | 225.7 | 61.9 | C = 1.29 E-02; Nitri = 7.18 E-05 |
| | ~ s(C) +s(Lat) | 226.0 | 61.9 | C = 2.24 E-02; Lat = 2.25 E-05 |
| | ~ s(C) +s(OM) | 224.1 | 55.3 | C = 3.97 E-03; MO = 1.71 E-04 |
| | ~ s(C) +s(Temp) | 231.4 | 55.8 | C = 9.6 E-04; Temp = 4.1 E-04 |
| | ~ s(C) +s(Lon) | 234.2 | 49.5 | C = 1.11 E-04; Lon = 6.64 E-03 |
| | ~ s(C) | 239.8 | 33.1 | C = 2.59 E-04 |

**Analyses combination used**: s(C); s(C) +s(Lat); s(C) +s(Lon); s(C) +s(Temp); s(C) +s(pH); s(C) +s(DO); s(C) +s(Nitra); s(C) +s(P); s(C) +s(Nitri); s(C) +s(MO); s(Nitra); s(Nitri).

**Acronyms:** s() = smoothness, C = conductivity, Lat = latitude, P = phosphate, S = silicic acid, OM = Organic Matter, Nitri = nitrite Temp = temperature, DO = dissolved oxygen.

Env—FG and FG—GHG presented significant (p <0.001) linear correlations $r^2$> 0.75 (Fig 6, S7 Table). The SEM model was supported by a good prediction, based on Chi-square ($X^2$, p = 0.301) and the model fit indexes obtained (i.e., RMSEA = 0.058; TLI = 0.965 and CFI = 0.971 (R output is summarized in S7 Table)).

## 4. Discussion

Coastal wetlands in the semiarid region of central-northern Chile are characterized by heterogeneous physical and chemical conditions in waters and sediments across latitude and season.

**Table 3. GAM models using environmental factors with nutrients as predictor variables and functional groups as response variables.**

| GHG | Models | AIC | ED | p-value |
|---|---|---|---|---|
| *Betaproteobacteria* | ~ s(C) +s(OM) | 563.4 | 67.2 | C < 2 E-16; MO = 2.56 E-02 |
| | ~ s(C) +s(P) | 596.2 | 63.8 | C < 2 E-16; P = 4.55 E-03 |
| | ~ s(C) + s(Lat) | 599.0 | 60.0 | C < 2 E-16; Lat = 4.44 E-02 |
| | ~ s(C) | 601.1 | 55.7 | C < 2 E-16 |
| *Methylotenera* sp | ~ s(C) +s(Nitri) | 229.3 | 52.6 | C = 1.56 E-04; Nitri = 8.06 E-03 |
| | ~ s(C) | 343.1 | 20.0 | C = 1.08 E-03 |
| *Nitrosomonas* sp | ~ s(C) | 394.8 | 28.5 | C = 6.73 E-05 |
| *Nitrospira* sp | ~ s(C) +s(Temp) | 486.4 | 39.1 | C = 9.82 E-03; Temp = 3.53 E-0 |
| | ~ s(C) +s(P) | 487.5 | 29.8 | C = 3.86 E-03; P = 7.51 E-02 |
| | ~ s(C) +s(DO) | 487.4 | 24.4 | C = 1.42 E-04; DO = 5.04 E-02 |
| | ~ s(C) | 488.0 | 19.2 | C = 1.74 E-03 |

**Analyses combination used:** s(C); s(C) +s(Lat); s(C) +s(P); s(C) +s(MO); s(C) +s(Nitri); s(C) +s(Temp); s(C) +s(DO).

**Acronyms:** s() = smoothness, C = conductivity, Lat = latitude, P = phosphate, S = silicic acid, OM = Organic Matter, Nitri = nitrite Temp = temperature, DO = dissolved oxygen.

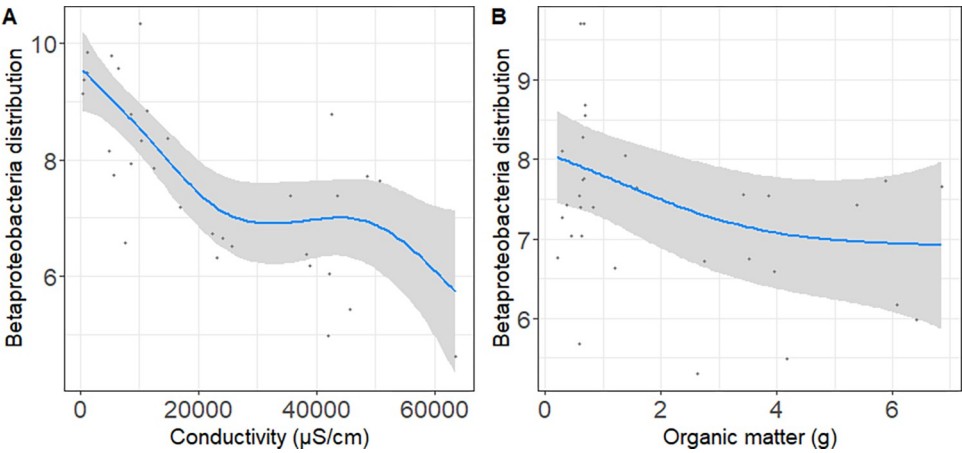

**Fig 5.** GAM models showing the relationship between the Betaproteobacteria changes with conductivity (A) and organic matter (B) as predictors. Axes "Y" correspond to the effect of the different predictor variables.

Indeed, our study revealed that hyposaline waters and coarse-gravel sediments are more frequently observed in the southern basins, belonging to the low aridity basins (LAB). In contrast, mesosaline water conditions and very fine-sand sediments with higher organic matter content were mostly found in the northern basin´s wetlands, belonging to the high aridity basins (HAB) (Fig 2A and 2B, S3A and S3B Fig). These results are consistent with the geographic location of the wetlands and landscapes scales, associated with aridity shifts from north to south basins. Our study found that LAB wetlands belonging to southern basins were characterized by high nutrient enrichments (Fig 2, S1 Table), which could be potentially associated with a greater anthropogenic pressure from urban areas and greater utilization of water for industry and agriculture in the southern respect to their northern counterpart. This is in agreement with previous reports that showed the significant impact of irrigated agriculture on N-content in the Aconcagua watershed [46]. In addition, various pollution sources, besides domestic water, were reported in Laguna Verde, a wetland categorized for having the worst state of conservation compared with other wetlands from central-northern Chile [32], including water discharge from sewage treatment plants and percolation of municipal landfill [47]. Moreover, the central-northern area of Chile is experimenting a strong climate change scenario including a megadrought [29], that influences the water availability and quality in the basins [48], and potential variable salinization mechanisms, besides seawater intrusion, as observed in other coastal wetlands [30].

## 4.1 Dissolved greenhouse gases and potential regulation by environmental factors associated with latitude changes HAB to LAB

The results of dissolved greenhouse gasses (GHG) distribution indicated that the coastal wetlands situated at LAB presented higher concentration compared with their counterparts from HAB, resulting in a greater GHG excess estimation, which indicates the potential of water-atmosphere exchange. As in other estuarine wetlands a positive $CH_4$ excess was registered, whereas $CO_2$ was more variable, showing a high excess magnitude in winter at the LAB area (Fig 3). On the other hand, $N_2O$ excess indicated a clear latitudinal trend, as HAB wetlands were potential $N_2O$ sinks, while LAB wetlands were potential sources of $N_2O$ to the atmosphere in summer. We are aware that the results of analyses related to GHG excess can be considered as a potential trend, considering that only a single point of each estuarine wetland was

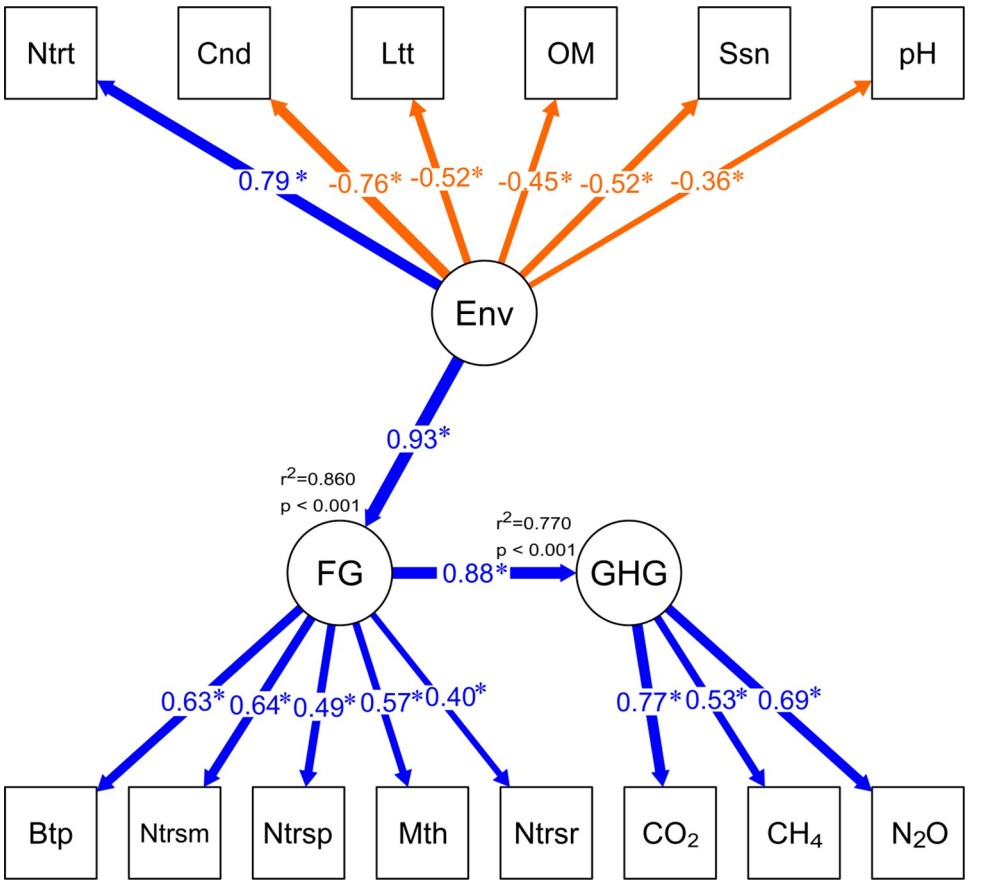

**Fig 6. Structural equation model (SEM) showing the influences of observed factors on latent variables and the identification of the effect of the environment (ENV) on functional microbial groups (FG) and the FG influence on Greenhouse gases (GHG).** Arrows represent the relationships between observed factors and latent. Significance was signed with an asterisk (<0.05) and path coefficients values were included for latent potential causality effects.

studied, and samples were exclusively associated with water. Moreover, a great diel variability is expected in the water and vegetation of coastal wetlands, as reported in other wetlands associated with *P. australis*, one of the most prevalent aquatic plants inhabiting estuaries around the world [5]. In our study sites, a greater influence of vegetation is expected in the wetlands located in LAB (see pictures S1 and S2 Figs). Future studies should consider flux estimation approaches that include sediments, water and surrounding biomes to identify net GHG contribution of the wetlands in central-northern Chile and corroborate our baseline results.

Our GAM models indicate that conductivity was a common factor predicting $CO_2$ and $CH_4$ variability (Fig 4, Table 2). However, the variability of the prediction for $CH_4$ and $CO_2$ increased when considering latitude and pH, respectively (Fig 4A–4D; Table 2). On the other hand, latitude and nitrate were the two best predictors accounting for the distribution of $N_2O$ concentrations and not conductivity. Our results are consistent with previous studies in temperate areas considering the role of salinity combined with the effect of nutrients in some wetlands. For example, a decrease in $CO_2$ and $CH_4$ effluxes in the Yellow River Delta wetland [49] and the Yangtze Estuary [50], associated with higher salinity, water depth and tide cycles [50–52].

On the other hand, experimental studies combining the effect of salinity and nutrients enrichment in freshwater wetlands have reported that independent of salinity, $N_2O$ increment and reduction/suppression of $CH_4$ production were associated with nitrate inputs [53].

Therefore, nutrients play an important role in the dynamics of GHG in estuaries and wetlands. In the study area, potential latitudinal changes in the role of microbial processes involved in nitrogen recycling, such as nitrification and denitrification are expected. Supporting this idea, we found that nitrate as a product of nitrification was significantly correlated with the $N_2O$ in the coastal wetlands (Fig 4E and 4F; S3 Table), as reported elsewhere for constructed wetlands [54]. Moreover, our correlations suggest a greater potential role of nitrification in the $N_2O$ accumulation in LAB wetlands compared with HAB. Likewise, reads assigned to ammonia oxidizing bacteria and archaea were correlated with $N_2O$ and $NO_3^-$ (e.g., *Nitrosomonas* and *Nitrosoarchaeum*, and other genera S6 Table). In addition, the predominant $N_2O$ sink estimated in the HAB wetlands potentially indicates an active consumption that could be associated with anaerobic processes, such as denitrification.

## 4.2 Microbial community structure and key functional groups potentially associated with GHG regulation

The benthic microbial community associated with SAZCCh coastal wetlands was found to be diverse and exhibited changes associated with environmental factors including nutrients, temperature and conductivity variability. For example, PCoA analysis revealed that at ASV level, the microbial community structure was predominantly grouped according to hyposaline and mesosaline conditions, with slight changes in their diversity (S4A and S4B Fig). Salinity is a well-known environmental factor modulating microbial community structure at global [55] and local [56] scales, such as estuaries [57]. In these transition areas, the microbial community structure changes along salinity gradients were associated with a greater stochastic input of microbes from marine habitats during the tidal regime [58] and by survival and adaptation of those microorganisms to freshwater environments [59].

Functional groups associated with GHG recycling were also retrieved in our 16S rRNA gene survey, including ammonia oxidizing archaea and bacteria, methanogens and methanotrophs, showing significant correlations with environmental variables including GHG and negative correlations with conductivity (S6 Table). GAM analyses support environmental factors such as conductivity, latitude and OM as predictor factors in the distribution of *Betaproteobacteria* (Table 3).

The high number of negative correlations with conductivity agrees with previous reports focused in functional groups. For instance, *Nitrosomonas* dominated the freshwater areas compared to *Nitrosospira* which was more abundant in brackish water of the Ythan estuary (Scotland) [60]. Previous research has documented that the abundance of other methanotrophic species, such as *Methylomirabilis oxifera* and *Methylocystis* sp., decreased in areas with higher salinity of coastal wetlands [61, 62].

Structural equation modeling (SEM) supports the importance of the environment (ENV) considering environmental factors, such as, conductivity, pH, OM, nutrients, geographical and temporality conditions sampled (latitude and season) as significant predictors for the selected functional microbial group (FG) conformed by five grouped selected taxa (Betaproteobactera, Nitrosomonas, Nitrospira, Methylotenera and Nitrosoarchaeum), which are further related with GHG as latent variable.

The ammonia oxidizer *Nitrosomonas* sp. presented a significant contribution to the variability within the FG using SEM modeling and also individually based in GAM associated with conductivity. This result supports reports associated with *Nitrosomonas* sp. as a predominant ammonia oxidizer in estuaries including anthropogenically perturbed wetlands [63]. Therefore, possibly responsible for the higher $N_2O$ concentration found in the LAB wetlands resulting in positive excess.

The potential role of functional groups associated with both methanotrophic, nitrification and denitrification pathways should be further studied to evaluate their role in $N_2O$ production. More adequate methods, such as specific quantification by RT-PCR, together with a higher spatiotemporal replicability, could help to generate more integrative analyses and model the contribution of key microbial groups in nutrients GHG dynamics, including potential latitudinal trends associated with a higher accumulation of $N_2O$ in the water of hyposaline wetlands situated at the southern basins.

## 5. Conclusion

Coastal wetlands in the semiarid zone of central-northern Chile are heterogeneous and very distinctive ecosystems characterized by a latitudinal variability of their physicochemical conditions and their benthic microbial community. The findings of this study indicate that salinity combined with different environmental factors depending on the specific GHG studied accounted for the dissolved greenhouse gas variability. The results revealed a latitudinal shift in the dissolved $N_2O$ distribution likely associated with the input of allochthonous nutrients probably associated with higher anthropogenic influence on these wetlands located in the south. Salinity and nutrients were factors associated with the composition of key functional groups of microorganisms, including methane-oxidizing, such as *Methylobacter* sp. and *Methylotenera* sp. Our study highlights how a set of highly diverse wetland microbiomes respond to the combined effect of salinity and nutrient input, contributing directly or indirectly to the regulation of the biogeochemistry of the wetlands. The dynamic of these coastal water bodies is becoming especially important to get a better understanding of the potential greenhouse gases emissions, where a greater effort is needed to integrate spatiotemporal variability of nutrients, GHG and microbial communities.

## Supporting information

**S1 Table. Physical, chemical and biological (functional groups and phyla) data determined during our study from a selected group of estuarine coastal wetlands of the central-northern Chile.**
(XLSX)

**S2 Table. Sequencing reads count information before and after denoise and links for repository download.**
(XLSX)

**S3 Table. Spearman multiple correlation analyses between physical and chemical water conditions, nutrients and greenhouse gases distribution, sediment organic matter content and granulometry.** The values show the correlation coefficient (r) and the three colors the significance level of p-value.
(XLSX)

**S4 Table. Alpha -diversity indexes (richness, Shannon index and evenness) of the benthic microbial community from the coastal wetlands in winter and summer.**
(XLSX)

**S5 Table. SIMPER analyses associated with the comparison of microbial communities contributing >5% associated with the different categories analyzed, i.e., conductivity, latitude and season sampled.**
(XLSX)

**S6 Table. Spearman multiple correlation analyses between physical-chemical variables (environmental factors and nutrients) and the functional group.** The green taxa were selected for the GAM analyses. The values show the correlation coefficient (r) and the three colors the significance level of p-value.
(XLSX)

**S7 Table. Structural equation model (SEM) statistical summary results.**
(XLSX)

**S1 Fig. Coastal wetlands pictures sampled in the Coquimbo region.**
(TIF)

**S2 Fig. Coastal wetlands pictures sampled in the Valparaíso region.**
(TIF)

**S3 Fig.** Granulometry and organic matter in the sediments of the sampled wetlands in winter (A) and summer (B). Teatinos, Chigualoco, Cartagena and Yali wetlands were only measured in summer and Salinas G, La Cebada and Marga Marga wetlands were only measured in winter.
(TIF)

**S4 Fig.** A) Principal Coordinate Analysis in hyposaline and mesosaline wetlands. Acronyms in the wetland names are H = hyposaline, M = mesosaline, IV = Coquimbo Region, V = Valparaíso Region, W = Winter, V = Summer. B) Boxplot (median, quartiles, and outliers) showing grouped richness values considering the saline categories.
(TIF)

**S5 Fig.** Benthic bacterial phyla observed in winter (A) and summer (B). The Proteobacteria phylum was separated in the Gamma, Delta and Alphaproteobacteria classes. Low abundant Phyla (<0.1%, Acetothermia, Aegiribacteria, AncK6, Armatimonadetes, Atribacteria, BHI80-139, BRC1, CK-2C2-2, Caldiserica, Calditrichaeota, Chlamydiae, Cloacimonetes, Coprothermobacteraeota, Dadabacteria, Deferribacteres, Deinococcus-Thermus, Dependentiae, Elusimicrobia, Entotheonellaeota, Epsilonbacteraeota, FBP, FCPU426, Fibrobacteres, Fusobacteria, GAL15, Halanaerobiaeota, Hydrogenedentes, Kiritimatiellaeota, LCP-89, Latescibacteria, Lentisphaerae, MAT-CR-M4-B07, Margulisbacteria, Marinimicrobia (SAR406 clade), Modulibacteria, Nitrospinae, Omnitrophicaeota, PAUC34f, Patescibacteria, Poribacteria, Rokubacteria, Schekmanbacteria, Synergistetes, TA06, Thermotogae, WOR-1, WPS-2, WS1, WS2, WS4, Zixibacteria.
(TIF)

**S6 Fig.** Benthic archaea phyla retrieved in winter (A) and summer (B).
(TIF)

**S7 Fig.** A) Non-metric multidimensional scaling (NMDS) analysis for selected functional groups (methane oxidizing bacteria, ammonia oxidizing bacteria and archaea, and methanogens). B) Linear fit associated with the NMDS analysis.
(TIF)

## Acknowledgments

We acknowledge M. Contreras, C. Zuleta, V. Bravo and E. Bassi for their support in the field campaigns. We are grateful to H. Peña, C. Sepúlveda, P. Reinoso and A. Bello for their technical support in the laboratory for molecular, nutrient and GHG analyses. This is publication

No. 91 of the Sino-Australian Research Consortium for Coastal Management (previously the Sino-Australian Research Center for Coastal Management) and of the network RED21992 "Sistema articulado de investigación en Cambio Climático y Sustentabilidad en zonas costeras de Chile".

## Author Contributions

**Conceptualization:** Francisco Pozo-Solar, Julio Salcedo-Castro, Verónica Molina.

**Data curation:** Francisco Pozo-Solar, Marcela Cornejo-D´Ottone, Roberto Orellana, Daniela V. Yepsen, Nickolas Bassi, Polette Aguilar-Muñoz.

**Formal analysis:** Francisco Pozo-Solar, Polette Aguilar-Muñoz, Verónica Molina.

**Funding acquisition:** Marcela Cornejo-D´Ottone, Roberto Orellana, Julio Salcedo-Castro, Verónica Molina.

**Investigation:** Francisco Pozo-Solar, Marcela Cornejo-D´Ottone, Verónica Molina.

**Methodology:** Marcela Cornejo-D´Ottone, Roberto Orellana, Daniela V. Yepsen, Nickolas Bassi, Julio Salcedo-Castro, Polette Aguilar-Muñoz.

**Project administration:** Verónica Molina.

**Supervision:** Roberto Orellana, Verónica Molina.

**Validation:** Daniela V. Yepsen, Verónica Molina.

**Visualization:** Daniela V. Yepsen, Nickolas Bassi.

**Writing – original draft:** Francisco Pozo-Solar.

**Writing – review & editing:** Francisco Pozo-Solar, Marcela Cornejo-D´Ottone, Roberto Orellana, Daniela V. Yepsen, Julio Salcedo-Castro, Verónica Molina.

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
