## [Decision Letter · Decision Letter 0]

13 Jan 2022

PONE-D-21-30349Dissolved greenhouse gases and benthic microbial communities in coastal wetlands of a large latitudinal gradient in a semiarid regionPLOS ONE

Dear Dr. Molina,

Thank you for submitting your manuscript to PLOS ONE. After careful consideration, we feel that it has merit but does not fully meet PLOS ONE’s publication criteria as it currently stands. Therefore, we invite you to submit a revised version of the manuscript that addresses the points raised during the review process.

 I recommend that you carefully consider the comments of both reviewers when carrying out the major revisions required for this manuscript. Neither reviewer felt that the statistical analyses were sufficient and it will be critical to include the proper analyses in a revised manuscript. The data for the manuscript must also be made publicly available and easy to access (with links included in the manuscript). Both reviewers made detailed comments on grammatical shortcomings and awkward language in the manuscript, and these points should be addressed. Reviewers felt that the introduction could be streamlined and that there should be an inclusion of greater interpretation of the data and possible mechanisms included in the discussion. 

We look forward to receiving your revised manuscript.

Kind regards,

Theodore Raymond Muth

Academic Editor

PLOS ONE

Journal Requirements:

V.M. FONDECYT 1171324, 1211977 Fondo Nacional de Ciencia y Tecnología 

V.M. and R.O. PAI79170091  Programa de Inserción a la Academia

https://www.anid.cl/concursos/

NO

This study was supported by FONDECYT grants 1171324 and 1211977, the Proyecto de Tesis de Postgrado at UPLA for FP, the CONICYT – Programa de Inserción a la Academia Project “Fortalecimiento de la Investigación y la Docencia en las Áreas de Microbiología Ambiental y Bioinformática de la Universidad de Playa Ancha” PAI79170091 (VM and RO), and Project “Apoyo a la formación de Redes Internacionales para Investigadores en Etapa Inicial” 170600 (RO). D. V. Yepsen received funding from ANID-PFCHA/Doctorado Nacional/2017-21170986. 

V.M. FONDECYT 1171324, 1211977 Fondo Nacional de Ciencia y Tecnología 

V.M. and R.O. PAI79170091  Programa de Inserción a la Academia

https://www.anid.cl/concursos/

NO

6. We note that Figure S1 and S2 in your submission contain copyrighted images. All PLOS content is published under the Creative Commons Attribution License (CC BY 4.0), which means that the manuscript, images, and Supporting Information files will be freely available online, and any third party is permitted to access, download, copy, distribute, and use these materials in any way, even commercially, with proper attribution. For more information, see our copyright guidelines: http://journals.plos.org/plosone/s/licenses-and-copyright.

a. You may seek permission from the original copyright holder of Figure S1 and S2 to publish the content specifically under the CC BY 4.0 license. 

Reviewers' comments:

Reviewer's Responses to Questions

**Comments to the Author**

1. Is the manuscript technically sound, and do the data support the conclusions?

Reviewer #1: Yes

Reviewer #2: Yes

2. Has the statistical analysis been performed appropriately and rigorously? 

Reviewer #1: No

Reviewer #2: No

3. Have the authors made all data underlying the findings in their manuscript fully available?

Reviewer #1: Yes

Reviewer #2: No

4. Is the manuscript presented in an intelligible fashion and written in standard English?

Reviewer #1: Yes

Reviewer #2: Yes

5. Review Comments to the Author

Reviewer #1: In the manuscript "Dissolved greenhouse gases and benthic microbial communities in coastal wetlands of a large latitudinal gradient in a semiarid region" by Francisco Pozo-Solar and collaborators, the authors studied microbial communities and greenhouse gases in Chilean coastal wetlands. The introduction seemed too long, and at a time, it felt that some details might not be so relevant to highlight in the introduction.

"In this study, the influence of environmental factors on GHG concentration and benthic microbial community composition in wetlands distributed between 33'77'S and 29'82'S was determined." – It is better to say Chilean coast compared to coordinates because these coordinates alone give very little information of the location.

"GAM analysis also indicated that conductivity and nutrients accounted for the changes in the relative abundance (n° sequences) of Betaproteobacteria and Methylotenera sp." – What mean "n° sequences"? Why highlight here Betaproteobacteria and Methylotenera sp.?

Lines 274-297 – It seems that the amount of data not big enough for such complicated models.

Figures 2 and 3 – Similar areas should be grouped, and standard deviation or error can be shown. The detailed graph may be shown in the Supplementary materials. Figure 2 and 3 can be merged. These figures show more background information.

Why different colours for winter and summer in Figures 2, 3 and 4? Are they showing something different?

Figure 5 – Cleary, some of the models are not describing the data enough. In particular, I mean sections E and F.

Units are missing or not correct in many figures (see Figures 4, 5, 6).

In the case of Figure 6, there were only up to ten sequences? How can the number of sequences be negative?

Would you please provide additional data tables for the quality of the sequences? A table indicating the number of total reads and number of reads after quality control and denoising is needed in the Supplementary material to compare the sequencing efficiency and control for any differences in sampling depth.

To be accepted for publication in PLOS ONE, research articles must satisfy the following criteria:

1. The study presents the results of original research.

Yes, this is correct.

2. Results reported have not been published elsewhere.

Yes, this seems correct.

3. Experiments, statistics, and other analyses are performed to a high technical standard and are described in sufficient detail.

I liked that the authors had a detailed overview of statistics.

4. Conclusions are presented in an appropriate fashion and are supported by the data.

More or less. I have some concerns regarding the size of the data and analysis. It seems that some analyses are not suitable with so little number of observations.

5. The article is presented in an intelligible fashion and is written in standard English.

Yes, this is correct.

6. The research meets all applicable standards for the ethics of experimentation and research integrity.

Yes, this is correct.

7. The article adheres to appropriate reporting guidelines and community standards for data availability.

Yes, this is almost correct. Some details about sequencing are more needed.

Reviewer #2: The manuscript is interesting and presents important analysis, based on an extensive dataset.

I have 2 major comments:

1) The underlying data is not made fully available.

2) The statistical analysis has issues with cross-variation of effects, and multiple testing of the same dataset.

See more details about these two issues below, as well and several minor comments of a more editorial nature.

I do not like much the phrase “large latitudinal gradient in a semiarid region”. It is somewhat of an oxymoron. The southern semiarid belt is typically considered a narrow latitudinal belt between the tropic and the mid-latitude. And while what is “large” is subjective, this is really pushing the boundary on the common convention of what would be considered a large gradient in a global context. I suggest making the title and similar claims throughout more subjective, and more to the point, by using the phrase “coastal wetlands along a latitudinal gradient through a semiarid region”. (Dropping the “large”. You do not really need it).

There are many of places with awkward and ambiguous phrasing and sentence structures. Nothing detrimental, but a bit distracting. Specifically, sentences that mean that do not distinguish between the environment surrounding the wetlands and the wetlands themselves. For example (abstract): “Wetlands in the north were associated with a higher aridity and lower anthropogenic influence compared with wetlands in south” technically reads as the wetlands are arid, where the intent is actually to say that the environments surrounding the wetlands are along a north-south gradient of higher aridity and lower anthropogenic impact.

Similarly, sentence word order is awkward is places such as (abstract): “(GAM) indicated that conductivity accounted for the larger variability of CH4 and CO2, but the predictions were improved when latitude and pH for CH4 and CO2 concentration were included”. This confused me, as I wasn’t sure what “pH for CO2 and CH4 concentration” means (is there a separate pH for CO2 and CH4?). I think this was intended to be: (GAM) indicated that conductivity accounted for the larger variability of CH4 and CO2, but the predictions of CH4 and CO2 concentration were improved when latitude and pH for were included. (i.e., make a direct placement of the verb after the subject: predictability of CO2 CH4 was improved, latitude and pH were included).

This is throughout the entire paper. Not sure how to address it, other than let some English and editing savvy writer to do a deep review of the text. Again, not detrimental, but takes away from the reading experience.

The introduction section jumps back and forth between discussion of wetlands in general, and specific statements related to coastal wetlands. For example, the first sentence in the paper starts with “Wetlands are ecosystems that provide…” but a few words later, calls out “blue carbon”. I am not sure that all the wetlands you measured even qualify as blue carbon reservoirs (See https://www.thebluecarboninitiative.org/). Are your wetland sites all tidal? Are they mangrove? I propose to focus the introduction on coastal wetlands (no need for a general “what are wetlands” paragraph, as the first half of your second paragraph, but there is a strong need for more information about the global role of coastal wetlands).

L69-71: You state that “Currently, hydrodynamics is considered as the main regulator factor on GHG recycling in the water and sediment of estuarine wetlands, for example, by seawater intrusion through tides, and freshwater availability, associated with river discharge, precipitation and aquifer hydrology”. I do not think there is such a broad agreement that hydrodynamics is the main regulator on GHG fluxes. Indeed, hydrodynamics determine where is a wetland vs. where it is dryland, but in permanently flooded wetlands (as the ones you studied) temperature (see figure 5 in Delwich et al 2021 ESSD) and vegetation carbon uptake (which is the assumption that drives most global models such as ELM) often have as strong effect as water levels. Very specific to tidal wetlands, salinity and saltwater intrusion are important (somewhat related to but not exactly a hydrodynamic effect, in fact, you get to that 3 paragraphs below). There are many more references and many more general review papers that discuss in more details what are the drivers and controls of GHG emissions in different wetland types, and some of these could be referenced here. Please improve this section in the introduction.

L128 no need for hyphen in central Chile.

L134-137: I do not like the hypotheses section. It reads as a combination of the summary of the results with the general statement “there will be a difference”, which doesn’t count much as a hypothesis, because with enough variability there will always be some difference. Please form mechanistic and testable hypotheses: do you expect higher or lower GHG production with increasing aridity? With increasing anthropogenic influence? Why? How do any of these will affects exactly what in the microbial community? Do you expect more bacteria/archaea that participate in what metabolic pathway in wetlands with higher GHG? Why? Which environmental variable that change throughout the north-south gradient is the dominant driver of GHG concentrations, and why?

Methods- you provide a general description of the regions (somewhat boring, but can stay), but missing a general description of the wetlands (which is critical to this paper). There should be a subsection “study sites” that explains how many sampling locations (“wetlands”) you sampled, how many sample locations (only 1? multiple depths?) and times (I assume 1 per season) within each (this can be almost inferred from the current text, but not directly stated in one organized place). Describe the wetlands, are they all estuarine coastal? Marshes? Tidal influence? Mangrove? Macrophyte or submerged vegetation? Are they all permanently flooded, or seasonal? Is there a gradient of these properties as you go north? What is the gradient of annual mean (or seasonal mean) temperature throughout your wetlands? Precipitation? There is a lot that should be known about your sample sites that you do not provide. I suggest adding as much environmental information to your table S1 as possible. Specifically, wetland biome (mangrove, marsh…), dominant vegetation species, mean annual temperature (and temperature during the sampling), annual precipitation, mean water depth (or depth at sampling time and location), salinity (not only the salinity category), river discharge.

That brings me to one of my most critical points. Your publication should provide the underlying data, and there is great importance in making the data of your study available (it is also required by the data policy of PLOS One, as almost all journals). Please add the numbers rather than "x" in every location in Table S1. Some columns (such as GHG) will need to be expanded (GHG is actually 3 different gasses concentrations, and for GHG you can combine table S3 with S1, or keep it as a separate table. Same for "Nutrients", possibly other columns in S1). Are each of your numbers (in table S3, and everywhere else) based on a single sample? That is dangerous. Assuming that it is not, please report the sample size and the standard deviation in addition to the mean. The header of table S1 mentions “In situ P = TºC, Conductivity, pH, DO measured in situ, Nutrients”. Rather confusing, but in any case, where are the numbers for DO, pH, T, Conductivity? What does “Nutrients” mean? Please add multiple columns for the nutrients that you measure and provide the numbers. What is “Gran.”? What is “Seq.”? Are they components of BMC?

Tables of statistical results (S2, S4, S7), provide a very partial description of the results and are generally meaningless without the actual numbers the statistics are based on. As I ask above, please add all the environmental, nutrient and GHG numbers to table S1 (or, if you prefer, split to 4 tables for Environmental, GHG, nutrients, biodiversity). Each should include the sample size and std within each variable at each wetland and season. Please find a way to present the taxa numbers per wetland and season. Maybe a table with the density (or gene sequence numbers, or whatever number you are using to characterize the abundance of a specific ASV in a particular sample). That will make a lot of tables, but PLOS doesn’t limit you.

In the paper you say that “Sequences were deposited in the European Nucleotide Archive (ENA) under the project accession ID PRJEB44346 and the primary accession samples (ERR5725941-ERR5725957). This is great, and I am happy that you did that. But please, provide the links (or a reference to a published paper that include such links) that will allow the reader to access the data without looking too hard. For example, I found: https://www.ebi.ac.uk/ena/browser/view/ERR5725941?show=reads . Is that one of your datasets? If yes, it means that you need 17 such links, and you should put all of them in an appendix and reference the appendix from the methods section.

Statistical analysis: I am concerned by the effects of cross-correlations among your effects. For example, pH and T are highly correlated (0.658 in table S4, which I assume is the R^2. But can be the correlation coefficient, you do not say. Please provide units to EVERY number you show). What is the point of including both of them as independent effects in the GAM (lines 279, 289). I suggest trimming down your model and only including “independent” drivers, by excluding one of each pair of variables with a strong correlation (at least 0.4? the threshold is arbitrary but say what you chose). Or use a principal component analysis and include the first 2-4 components in the GAM instead of the environmental variables.

It will be interesting to add the GHGi variables to the model for Taxa (line 289). If you reduce some of the highly correlated environmental variables you will have enough degrees of freedom left to do so.

Table S2 – the way you construct the test here is wrong, and influenced by multiple testing. For each resulting variable (environmental factors, GHG, Nutrients, OM) make a single test (statistical model) using Region and Season as effects (driving variables). Currently you make 5 assumingly independent tests. Because it is a single dataset (the same numbers participate in more than one test) your current assumed degree of freedom in these “independent” tests is wrong. Also reduce the number of models. Because we already know that, for example, pH and Temperature are strongly correlated, there is nothing to learn from a model of pH after we see the results of the model for Temperature.

6. PLOS authors have the option to publish the peer review history of their article (what does this mean?). If published, this will include your full peer review and any attached files.

Reviewer #1: No

Reviewer #2: No

---

## [Author Response · Author response to Decision Letter 0]

7 Apr 2022

R: Formatting sample was used as template in the new manuscript submission.

R: An statement was added in the Methods section indicating which organisms authorized field site access. 

R: We check and kindly ask to include additional financial support: 

V.M. FONDECYT 1171324, 1211977 Fondo Nacional de Ciencia y Tecnología 

V.M. and R.O. PAI79170091 Programa de Inserción a la Academia

https://www.anid.cl/concursos/ “ JS-C. Ministerio del Medio Ambiente (Licitación Pública N° 608897-43-LP81 “Determinación del riesgo de los impactos del Cambio Climático en las costas de Chile” and Centro de Cambio Global UC. FP. Proyecto de Tesis de Postgrado at UPLA. R.O. Grant “Apoyo a la formación de Redes Internacionales para Investigadores en Etapa Inicial” 170600. D.Y. ANID-PFCHA/Doctorado Nacional/2017-21170986. FONDEQUIP EQM160131

In addition, please add the suggested statement by the journal: 

R: The acknowledgment section was rephrased separated from financial disclosure.

R. Figure 1 was originally generated by one of our authors N.Bassi, using free available cartography softwares based on Shapefiles provided by different governmental institutions. A brief statement was included in the figure legend. Briefly, the map contains two layers. The first one is the “basins limits” that are freely available from the Ministerio de Obras Públicas through the Water Resources Directorate (Dirección General de Aguas, DGA) in the following link: http://catalogo.geoportal.cl/geoportal/catalog/search/resource/resumen.page?uuid=%7B7CBD895E-D0AC-48D7-886D-F5C3C99E6551%7D. and shapefile for public use in the following link: http://www.dga.cl/estudiospublicaciones/mapoteca/Cuencas/Limite_Cuencas_BNA.zip.

The second layer consists in the “administrative division” (División Política Administrativa 2020) provided by “Subsecretaría de Desarrollo Regional y Administrativo (SUBDERE)”, and publicly available in the following link: http://catalogo.geoportal.cl/geoportal/catalog/search/resource/resumen.page?uuid=%7BCFF07BC2-103F-42CD-B4CB-0AE4C5C6A9A4%7D and can be downloaded in the shapefile in this link: http://www.geoportal.cl/arcgis/rest/services/MinisteriodeInterior/chile_minterior_subdere_DPA2020/MapServer?f=lyr&v=9.3.

Permits were attached considering original Spanish and translations, besides copyrights permission provided by Nickolas Bassi who created this original figure.

The geographic coordinates of the wetlands were retrieved using GPS equipment (Garmin GPSMap 64S) allowing to include each wetland geographical position in the map using QGIS, a free geographic information system (GIS) open code version 2.18 version software and installation manuals (https://qgis.org/downloads/;
https://docs.qgis.org/2.18/es/docs/user_manual/preamble/features.html available in https://docs.qgis.org/2.18/es/docs/user_manual/introduction/getting_started.html#installation). Wetland positions were depicted with different colors considering the sampling season (Summer/Winter), where yellow and blue correspond to Summer and Winter, respectively. Circles with mixed colors represent that wetlands were sampled during both seasons.

6. We note that Figure S1 and S2 in your submission contain copyrighted images. All PLOS content is published under the Creative Commons Attribution License (CC BY 4.0), which means that the manuscript, images, and Supporting Information files will be freely available online, and any third party is permitted to access, download, copy, distribute, and use these materials in any way, even commercially, with proper attribution. For more information, see our copyright guidelines: http://journals.plos.org/plosone/s/licenses-and-copyright.

R. Figure S1 and S2 consists of pictures that were taken in the field by one of our authors, N.Bassi. He provided a permission letter.

Herein we attach the rebuttal letter, specifying the responses and improvements made thanks for the reviewer comments.

Veronica Molina

Reviewers' comments:

Reviewer's Responses to Questions

1. Is the manuscript technically sound, and do the data support the conclusions?

Reviewer #1: Yes

Reviewer #2: Yes

R.1. We review our manuscript to clarify more our scientific questions, results and highlight our findings in the conclusion.________________________________________

2. Has the statistical analysis been performed appropriately and rigorously?

Reviewer #1: No

Reviewer #2: No

R2. All statistical analyses were checked and new analyses were performed to gain better understanding of our results and provide adequate support to our conclusions. In the specific request generated by the reviewers, we provide detail information about the statistical overview (please check answer to comment 10) of analysis made for both, previous and current versions of this manuscript. 

3. Have the authors made all data underlying the findings in their manuscript fully available?

Reviewer #1: Yes

Reviewer #2: No

R.3. In the reviewed manuscript we provide a table (Table S1) consisting of all the data retrieved supporting our findings. Sequencing data were already uploaded to public repositories.

4. Is the manuscript presented in an intelligible fashion and written in standard English?

Reviewer #1: Yes

Reviewer #2: Yes

R.4. English grammar as well as the order of several paragraphs were checked and improved across all sections of the revised manuscript.________________________________________

5. Review Comments to the Author

REBUTTAL LETTER

Reviewer #1: 

1. In the manuscript "Dissolved greenhouse gases and benthic microbial communities in coastal wetlands of a large latitudinal gradient in a semiarid region" by Francisco Pozo-Solar and collaborators, the authors studied microbial communities and greenhouse gases in Chilean coastal wetlands. The introduction seemed too long, and at a time, it felt that some details might not be so relevant to highlight in the introduction.

R1. As commented by the reviewer, we agreed that the Introduction was not clear enough to shed light on the need to explore the relationship between environmental factors, GHG concentration and benthic microbial community composition in coastal wetlands along the coast of the semiarid region of Chile. The new version of the manuscript includes a better organized and streamlined version of the introduction, as well as new citations that highlighted the contribution of this study. 

2. "In this study, the influence of environmental factors on GHG concentration and benthic microbial community composition in wetlands distributed between 33'77'S and 29'82'S was determined." – It is better to say Chilean coast compared to coordinates because these coordinates alone give very little information of the location.

R2. We thanked the reviewer for this observation. We changed the statement in the abstract (lines 34-36) to “In this study, we evaluated the influence of environmental factors on GHG concentration and benthic microbial community composition in wetlands distributed along the coast of the semiarid region”. Taking this into account we further changed the title of the manuscript from “Dissolved greenhouse gases and benthic microbial communities in coastal wetlands of a large latitudinal gradient in a semiarid region” to “Dissolved greenhouse gases and benthic microbial communities in coastal wetlands of the Chilean coast semiarid region”

3. "GAM analysis also indicated that conductivity and nutrients accounted for the changes in the relative abundance (n° sequences) of Betaproteobacteria and Methylotenera sp." – What mean "n° sequences"? Why highlight here Betaproteobacteria and Methylotenera sp.?

R3. As we relegated the results of GAM analysis, this section of the abstract was removed. In this new version of the manuscript, the abstract highlights the result of the SEM “Structural Equation Modeling” analyses that help us to identify the connection between the environment, functional microbial groups and their resulting effect in “GHG” as latent variables, avoiding focusing only in a single or two microbial taxa. The reasons behind these changes are given in comment 10. 

4. Lines 274-297 – It seems that the amount of data not big enough for such complicated models.

R4. In agreement with the reviewer, we acknowledge that our study suffers the traditional challenges of any exploratory project in a relatively large set of wetlands (24), in which some were sampled in two seasons. Constrained by the tradeoff between sample size and information completeness per wetland, we choose the first one, as this study is the first survey applied along coastal wetlands belonging to SAZCCh. In addition, this study is also part of a project that will further focus on GHG dynamics of specific wetlands. 

Although the amount of data was not that big, it was sufficient to run GAM modeling and identify prediction variables and factors, mainly associated with the predominant order Betaproteobacteria, within the Gammaproteobacteria class, an order that holds a significant number of functional groups associated with GHG identified in our sequencing reads.However, considering both reviewers' suggestions different statistical approaches were used to identify relationships between the environment, functional microbial and the resulting GHG. Therefore, a Structural Equation Modeling (SEM) statistical analysis was proposed based in the causality power to test hypotheses in other ecological studies including soil microbiology studies , i.e., Dai et al., 2019 (Dai, Z., Liu, G., Chen, H. et al. Long-term nutrient inputs shift soil microbial functional profiles of phosphorus cycling in diverse agroecosystems. ISME J 14, 757–770 (2020). https://doi.org/10.1038/s41396-019-0567-9)

5. .Figures 2 and 3 – Similar areas should be grouped, and standard deviation or error can be shown. The detailed graph may be shown in the Supplementary materials. Figure 2 and 3 can be merged. These figures show more background information

R5. We thank the reviewer for this comment, because we agreed that this is a great idea to integrate both physical and chemical conditions and dissolved nutrients data per two groups of wetlands with contrasting characteristics, such as HAB (wetlands from high aridity basins) and LAB (wetlands from high aridity basins). In consequence, we have merged Figure 2 and Figure 3 of the last version of the manuscript into Figure 2 of the new version “ Physical and chemical conditions and dissolved nutrients determined in wetlands water grouped according to high versus low aridity basins and sampling period summer and winter. Outliers are indicated as black dots. The detailed table containing data for each wetland can be found in Table S1.”

6. Why different colours for winter and summer in Figures 2, 3 and 4? Are they showing something different?

R6: Figure 2 and 3 were replaced following the previous comment. Figure 4 depicts the season when GHG was sampled, since for Valparaíso GHG data are available for winter and summer. Details can be found in Table S1. 

7. Figure 5 – Cleary, some of the models are not describing the data enough. In particular, I mean sections E and F.

R.7. We thank the reviewer for this comment, since we ran the models again, we found some mistakes associated with nitrous oxide modeling. After all results were carefully checked, new analyses indicated that latitude and nitrate were significant predictors, therefore the figure 5 was changed.

8. Units are missing or not correct in many figures (see Figures 4, 5, 6).

In the case of Figure 6, there were only up to ten sequences? How can the number of sequences be negative?

R8. This was a mistake, we are sorry for this. The graph was edited considering the predicted variable in the axis which are the effect and not the exact unit value. 

9. Would you please provide additional data tables for the quality of the sequences? A table indicating the number of total reads and number of reads after quality control and denoising is needed in the Supplementary material to compare the sequencing efficiency and control for any differences in sampling depth.

R.9. Table S2 was provided in the new manuscript version indicating the sequencing analyses results. 

10. I liked that the authors had a detailed overview of statistics.

R.10. A detailed response is provided here and supported by two figures attached. The first version of the manuscript included an initial step to evaluate the relation between environmental factors (EF), nutrients (Ntr), organic matter (OM) and granulometry (Gran) on GHG and the contribution of functional groups (FG) based on a Spearman's multiple correlation analysis (Figure 1). According to the dispersion of the data of each factor, either t-Test or Mann Whitney-Test, was performed to evaluate if there were significant spatial and seasonal differences among EF, Ntr and OM. After this, a Non-Metric Multidimensional Scaling (NMDS) test was conducted using the relative abundance of functional groups to determine the influence of the season (winter versus summer) as a factor. The NMDS was used prior to the application of the Generalized Additive Model (GAM) in order to include all the winter and summer data together, avoiding the effect of season in data related to microbial communities. In other words, the NMDS showed that the winter and summer data populations did not cluster separately, and therefore, they could all be input together into the GAM models, the final stage of the statistical analysis. This last stage consisted of performing GAM models to evaluate the effect of conductivity and other environmental variables on GHG and selected functional groups variability. 

A couple of changes have been incorporated in the current version of the manuscript, which are shown in orange (Figure 2). The first change involved the replacement of both t-Test or Mann Whitney-Test by a 2-way ANOVA due to the fact that both methods were performed incorrectly, as data was wrongly grouped. We thanked reviewers for sparking this revision, since this allowed us to correct this error prior to publication of this manuscript. Therefore, we decided to do a single Two-Way ANOVA test to see if the above mentioned variables show spatial and seasonal differences. The second change in this revised version of the manuscript involved tha addition of a Structural Equation Modeling (SEM) that was applied with conductivity, pH, latitude, season, nitrate and organic matter as independent variables. SEM is an analysis that relates causality between selected independent variables and response variables through linear models and the incorporation of latent variables. Indeed, the idea was to evaluate how environmental factors may influence functional groups and those functional groups could influence GHGs. Since, this causal relationship cannot be stated using GAM, we believe this is a substantial improvement of the current version of this manuscript. 

Figure 1: Overview of statistical analyses that were included in the first version of this manuscript. Abbreviations are as following: EF, environmental factors; Ntr, concentration of nutrients; GHG, concentration of GHG, FG, functional groups including nitrifiers, methanogens and methanotrophs; OM, organic matter; Gran, granulometry; Lat, latitude; Lon, longitude; GAM, generalized additive model; NMDS, non metric multidimensional scaling. 

Figure 2: Overview of statistical analyses that were included in the new version of this manuscript. Abbreviations are as following: EF, environmental factors; Ntr, concentration of nutrients; GHG, concentracion of GHG, FG, functional groups including nitrifiers, methanogens and methanotrophs; OM, organic matter; Gran, granulometry; Lat, latitude; Lon, longitude; Sea, season; Nitra, nitrate; Cond, conductivity; GAM, generalized additive model; NMDS, non metric multidimensional scaling. 

11. Conclusions are more or less an appropriate fashion and are supported by the data. I have some concerns regarding the size of the data and analysis. It seems that some analyses are not suitable with so little number of observations.

R.11. As we stated above, we acknowledge that our study suffers from not having a more complete set of data per wetland. We agree with the reviewer that some readers may be disappointed to find this. However, we feel that a major effort has been taken to improve the quality of our manuscript taking in account the resources and time available to this research. Considering our new statistical approaches and a better and more detailed review of our procedures based on the valuable comments of both reviewers, the conclusion was sustained considering the main findings, which are statistically supported. These changes have helped us to identify main environmental factors influencing coastal wetlands GHG to support and guide also future studies associated specifically into one or selected estuaries.

To be accepted for publication in PLOS ONE, research articles must satisfy the following criteria:

1. The study presents the results of original research.

Yes, this is correct.

2. Results reported have not been published elsewhere.

Yes, this seems correct.

3. Experiments, statistics, and other analyses are performed to a high technical standard and are described in sufficient detail.

I liked that the authors had a detailed overview of statistics.

R.3 A detailed overview was included in the rebuttal letter and reformulated, including new approaches.

4. Conclusions are presented in an appropriate fashion and are supported by the data.

More or less. I have some concerns regarding the size of the data and analysis. It seems that some analyses are not suitable with so little number of observations.

R.4. As we agree with the reviewer, we have made changes to address these challenges. Those are explained along with the answers R4, R10 and R11. 

5. The article is presented in an intelligible fashion and is written in standard English.

Yes, this is correct.

6. The research meets all applicable standards for the ethics of experimentation and research integrity.

Yes, this is correct.

7. The article adheres to appropriate reporting guidelines and community standards for data availability.

Yes, this is almost correct. Some details about sequencing are more needed. 

R7. We add a table to make all the data available and sequencing information.

Reviewer #2: The manuscript is interesting and presents important analysis, based on an extensive dataset.

12. The underlying data is not made fully available.

R 12: A new supplementary table 1 was included in the new manuscript version with all the data available and used in our study. 

13. The statistical analysis has issues with cross-variation of effects, and multiple testing of the same dataset.

R13: We appreciate the comment however we don't fully understand if you are referring to cross correlation effects or something else. Considering the second part of your comment, we did perform multiple testing aiming to address different questions. For example, to compare the variability of environmental variables associated with high (HAB) versus low aridity basins (LAB) wetlands across different seasons. A second set of tests was aimed to evaluate physical and chemical variables correlating (or co-correlating) with greenhouse gases and functional microbial groups in order to predict the distribution of GHG. A detailed overview associated with statistical analyses was included in answer number 10. 

See more details about these two issues below, as well as several minor comments of a more editorial nature.

14. I do not like much the phrase “large latitudinal gradient in a semiarid region”. It is somewhat of an oxymoron. The southern semiarid belt is typically considered a narrow latitudinal belt between the tropic and the mid-latitude. And while what is “large” is subjective, this is really pushing the boundary on the common convention of what would be considered a large gradient in a global context. I suggest making the title and similar claims throughout more subjective, and more to the point, by using the phrase “coastal wetlands along a latitudinal gradient through a semiarid region”. (Dropping the “large”. You do not really need it).

R.14 Thanks for your suggestion, we agree. This was changed in the title of the manuscript from “Dissolved greenhouse gases and benthic microbial communities in coastal wetlands of a large latitudinal gradient in a semiarid region” to “Dissolved greenhouse gases and benthic microbial communities in coastal wetlands of the Chilean coast semiarid region”, also in other parts of the manuscript.

15. There are many of places with awkward and ambiguous phrasing and sentence structures. Nothing detrimental, but a bit distracting. Specifically, sentences that mean that do not distinguish between the environment surrounding the wetlands and the wetlands themselves. For example (abstract): “Wetlands in the north were associated with a higher aridity and lower anthropogenic influence compared with wetlands in south” technically reads as the wetlands are arid, where the intent is actually to say that the environments surrounding the wetlands are along a north-south gradient of higher aridity and lower anthropogenic impact.

Similarly, sentence word order is awkward is places such as (abstract): “(GAM) indicated that conductivity accounted for the larger variability of CH4 and CO2, but the predictions were improved when latitude and pH for CH4 and CO2 concentration were included”. This confused me, as I wasn’t sure what “pH for CO2 and CH4 concentration” means (is there a separate pH for CO2 and CH4?). I think this was intended to be: (GAM) indicated that conductivity accounted for the larger variability of CH4 and CO2, but the predictions of CH4 and CO2 concentration were improved when latitude and pH for were included. (i.e., make a direct placement of the verb after the subject: predictability of CO2 CH4 was improved, latitude and pH were included).

This is throughout the entire paper. Not sure how to address it, other than let some English and editing savvy writer to do a deep review of the text. Again, not detrimental, but takes away from the reading experience.

R.15. We are sorry for these grammatical errors and confusing ideas across the first version of the manuscript. We have deeply checked the manuscript and corrected the English for clarity avoiding odd phrasing in the second version.

16. The introduction section jumps back and forth between discussion of wetlands in general, and specific statements related to coastal wetlands. For example, the first sentence in the paper starts with “Wetlands are ecosystems that provide…” but a few words later, calls out “blue carbon”. I am not sure that all the wetlands you measured even qualify as blue carbon reservoirs (See https://www.thebluecarboninitiative.org/). Are your wetland sites all tidal? Are they mangrove? I propose to focus the introduction on coastal wetlands (no need for a general “what are wetlands” paragraph, as the first half of your second paragraph, but there is a strong need for more information about the global role of coastal wetlands).

R.16. The introduction section was entirely changed considering the suggestions and also reviewer 1 comments. The wetlands sampled are estuarine from temperate areas, not mangrove, therefore this paragraph was rephrased.

17. L69-71: You state that “Currently, hydrodynamics is considered as the main regulator factor on GHG recycling in the water and sediment of estuarine wetlands, for example, by seawater intrusion through tides, and freshwater availability, associated with river discharge, precipitation and aquifer hydrology”. I do not think there is such a broad agreement that hydrodynamics is the main regulator on GHG fluxes. Indeed, hydrodynamics determine where is a wetland vs. where it is dryland, but in permanently flooded wetlands (as the ones you studied) temperature (see figure 5 in Delwich et al 2021 ESSD) and vegetation carbon uptake (which is the assumption that drives most global models such as ELM) often have as strong effect as water levels. Very specific to tidal wetlands, salinity and saltwater intrusion are important (somewhat related to but not exactly a hydrodynamic effect, in fact, you get to that 3 paragraphs below). There are many more references and many more general review papers that discuss in more details what are the drivers and controls of GHG emissions in different wetland types, and some of these could be referenced here. Please improve this section in the introduction.

R17. We thank the reviewer comment and reference shared regarding the main drivers of GHG recycling in estuarine coastal wetlands.The introduction section was rewritten, considering your comment and the first reviewer, to streamline it and add background to sustain the hypotheses. 

18. L128 no need for hyphen in central Chile.

R.18. This was changed to central-northern Chile.

19. L134-137: I do not like the hypotheses section. It reads as a combination of the summary of the results with the general statement “there will be a difference”, which doesn’t count much as a hypothesis, because with enough variability there will always be some difference. Please form mechanistic and testable hypotheses: do you expect higher or lower GHG production with increasing aridity? With increasing anthropogenic influence? Why? How do any of these will affects exactly what in the microbial community? Do you expect more bacteria/archaea that participate in what metabolic pathway in wetlands with higher GHG? Why? Which environmental variable that change throughout the north-south gradient is the dominant driver of GHG concentrations, and why?

R.19. We agree that the prediction was missing, therefore the hypotheses was rephrased to “We hypothesize that salinity and latitudinal changes associated with water availability and nutrients enrichments, related with more anthropogenically influenced wetlands (polluted or urban), will increase dissolved GHG inventories by altering the contribution of nitrifiers, methanogens and methanotrophs in the benthic microbial communities. ”

20. Methods- you provide a general description of the regions (somewhat boring, but can stay), but missing a general description of the wetlands (which is critical to this paper). There should be a subsection “study sites” that explains how many sampling locations (“wetlands”) you sampled, how many sample locations (only 1? multiple depths?) and times (I assume 1 per season) within each (this can be almost inferred from the current text, but not directly stated in one organized place). Describe the wetlands, are they all estuarine coastal? Marshes? Tidal influence? Mangrove? Macrophyte or submerged vegetation? Are they all permanently flooded, or seasonal? Is there a gradient of these properties as you go north? What is the gradient of annual mean (or seasonal mean) temperature throughout your wetlands? Precipitation? There is a lot that should be known about your sample sites that you do not provide. I suggest adding as much environmental information to your table S1 as possible. Specifically, wetland biome (mangrove, marsh…), dominant vegetation species, mean annual temperature (and temperature during the sampling), annual precipitation, mean water depth (or depth at sampling time and location), salinity (not only the salinity category), river discharge.

R.20. A study area description was rephrased, the general description was deleted and more information regarding the wetlands were included in the first section of the Methods. The wetlands are coastal estuarine systems, permanently flooded systems, influenced by tides. Some are RAMSAR sites, like the case Tongoy, Salinas Chicas, Salinas Grande, Pachingo and Limarí in high aridity basins and only Yali at Low aridity basins. 

21. That brings me to one of my most critical points. Your publication should provide the underlying data, and there is great importance in making the data of your study available (it is also required by the data policy of PLOS One, as almost all journals). Please add the numbers rather than "x" in every location in Table S1. Some columns (such as GHG) will need to be expanded (GHG is actually 3 different gasses concentrations, and for GHG you can combine table S3 with S1, or keep it as a separate table. Same for "Nutrients", possibly other columns in S1). Are each of your numbers (in table S3, and everywhere else) based on a single sample? That is dangerous. Assuming that it is not, please report the sample size and the standard deviation in addition to the mean. The header of table S1 mentions “In situ P = TºC, Conductivity, pH, DO measured in situ, Nutrients”. Rather confusing, but in any case, where are the numbers for DO, pH, T, Conductivity? What does “Nutrients” mean? Please add multiple columns for the nutrients that you measure and provide the numbers. What is “Gran.”? What is “Seq.”? Are they components of BMC?

R 21: Thanks for the suggestion, In the new version of the manuscript, underlying data were added, including standard deviation for chemical analyses and GHG. A new supplementary table was constructed including all the available data.

22. Tables of statistical results (S2, S4, S7), provide a very partial description of the results and are generally meaningless without the actual numbers the statistics are based on. As I ask above, please add all the environmental, nutrient and GHG numbers to table S1 (or, if you prefer, split to 4 tables for Environmental, GHG, nutrients, biodiversity). Each should include the sample size and std within each variable at each wetland and season. Please find a way to present the taxa numbers per wetland and season. Maybe a table with the density (or gene sequence numbers, or whatever number you are using to characterize the abundance of a specific ASV in a particular sample). That will make a lot of tables, but PLOS doesn’t limit you.

R.22: Thanks for the suggestion, we agree that it is much better for clarity and for interested readers. We generated a full table showing all the data available and used for our statistical determinations as mentioned in the former response. In addition, frequency of the ASV associated with functional microbial groups were also included.

23. In the paper you say that “Sequences were deposited in the European Nucleotide Archive (ENA) under the project accession ID PRJEB44346 and the primary accession samples (ERR5725941-ERR5725957). This is great, and I am happy that you did that. But please, provide the links (or a reference to a published paper that include such links) that will allow the reader to access the data without looking too hard. For example, I found: https://www.ebi.ac.uk/ena/browser/view/ERR5725941?show=reads . Is that one of your datasets? If yes, it means that you need 17 such links, and you should put all of them in an appendix and reference the appendix from the methods section.

R.23. A new supplementary Table 2 was generated including the sequence quality analysis results, suggested also by Reviewer 1, with links to facilitate interested readers to download sequences from the repository ENA.

24. Statistical analysis: I am concerned by the effects of cross-correlations among your effects. For example, pH and T are highly correlated (0.658 in table S4, which I assume is the R^2. But can be the correlation coefficient, you do not say. Please provide units to EVERY number you show). What is the point of including both of them as independent effects in the GAM (lines 279, 289). I suggest trimming down your model and only including “independent” drivers, by excluding one of each pair of variables with a strong correlation (at least 0.4? the threshold is arbitrary but say what you chose). Or use a principal component analysis and include the first 2-4 components in the GAM instead of the environmental variables.

R.24. Spearman Rank analyses are given in Table S4 now. We checked our results and considered your suggestion avoiding cross-correlations in all the analyses performed considering a threshold limit R>0.5. For example, we didn't use temperature and pH, conductivity and silicate together to model the response of gases. 

The suggestion of using the principal component gave us an idea of incorporating a new analysis in our revised manuscript that can be used to explore latent variables derived from observed factors we refer to Structural equation modeling (SEM). This model was used to test the interaction of the environment, functional group and their resulting effect on greenhouse gases. An overview of the statistical methods were described in detail in R.10. 

25. It will be interesting to add the GHGi variables to the model for Taxa (line 289). If you reduce some of the highly correlated environmental variables you will have enough degrees of freedom left to do so.

R.25: Unfortunately, GHG and functional groups did not have the same resolution in our data set for GAM modeling, and since the functional group matrix having a better coverage in our wetlands represented four main groups, we considered them all in the SEM analyses, as explained above in the previous response. 

26. Table S2 – the way you construct the test here is wrong, and influenced by multiple testing. For each resulting variable (environmental factors, GHG, Nutrients, OM) make a single test (statistical model) using Region and Season as effects (driving variables). Currently you make 5 assumingly independent tests. Because it is a single dataset (the same numbers participate in more than one test) your current assumed degree of freedom in these “independent” tests is wrong. Also reduce the number of models. Because we already know that, for example, pH and Temperature are strongly correlated, there is nothing to learn from a model of pH after we see the results of the model for Temperature.

R26: We thank the reviewer for pointing this out and helping us to correct and improve our data analyses. Therefore, we used a single test (Two-Way ANOVA) using region and season as factors.

---

## [Decision Letter · Decision Letter 1]

6 Jun 2022

PONE-D-21-30349R1Dissolved greenhouse gases and benthic microbial communities in coastal wetlands of the Chilean coast semiarid regionPLOS ONE

Dear Dr. Molina,

Thank you for submitting your manuscript to PLOS ONE. After careful consideration, we feel that it has merit but does not fully meet PLOS ONE’s publication criteria as it currently stands. Therefore, we invite you to submit a revised version of the manuscript that addresses the points raised during the review process.

There remain several issues with the statistical analyses of the data in the revised version of the manuscript. Specifically, please address the reviewer's comments on Tables 2 and 3. Also be sure to address the comments on structural equation modeling. 

We look forward to receiving your revised manuscript.

Kind regards,

Theodore Raymond Muth

Academic Editor

PLOS ONE

Journal Requirements:

Reviewers' comments:

Reviewer's Responses to Questions

**Comments to the Author**

1. If the authors have adequately addressed your comments raised in a previous round of review and you feel that this manuscript is now acceptable for publication, you may indicate that here to bypass the “Comments to the Author” section, enter your conflict of interest statement in the “Confidential to Editor” section, and submit your "Accept" recommendation.

Reviewer #1: All comments have been addressed

2. Is the manuscript technically sound, and do the data support the conclusions?

Reviewer #1: Yes

3. Has the statistical analysis been performed appropriately and rigorously? 

Reviewer #1: No

4. Have the authors made all data underlying the findings in their manuscript fully available?

Reviewer #1: Yes

5. Is the manuscript presented in an intelligible fashion and written in standard English?

Reviewer #1: Yes

6. Review Comments to the Author

Reviewer #1: Thank you for this good work with your manuscript. It is good to see if the authors work hard to get their data and results publishable, even if there are restrictions in the case of data amount and some others, which may be, in some cases, out of our hands. Thus, I like the present version more compared with the earlier one.

The introduction has improved a lot. The results are now very much shown in a better way. The whole work starts to have a meaning. Thank you for that.

Still, some things regarding the statistics seem to be weird. First, the order of the models in tables 2 and 3 is not logical. Lower AIC values indicate a better-fit model; thus, the order should be changed to show better models at first.

Another problem may be with structural equation modelling (SEM). Structural equation modelling (SEM) aims to define a theoretical causal model consisting of a set of predicted covariances between variables and then test whether it is plausible when compared to the observed data. I cannot see what the basics for the model were; the initial theoretical causal model is missing. In addition, what are the overall parameters for the model to estimate the significance and correctness of the model? What were the p-value, root mean square error of approximation (RMSEA), comparative fit index (CFI), and Tucker–Lewis index (TLI)?

7. PLOS authors have the option to publish the peer review history of their article (what does this mean?). If published, this will include your full peer review and any attached files.

Reviewer #1: No

---

## [Author Response · Author response to Decision Letter 1]

19 Jun 2022

Herein we attach the rebuttal letter, specifying the responses and improvements made stated in the reviewer comments in section 6. In addition, we found some minor errors in the text which were listed and explained below highlighted in the text as requested.

6. Review Comments to the Author

(1) Reviewer #1: Thank you for this good work with your manuscript. It is good to see if the authors work hard to get their data and results publishable, even if there are restrictions in the case of data amount and some others, which may be, in some cases, out of our hands. Thus, I like the present version more compared with the earlier one.

The introduction has improved a lot. The results are now very much shown in a better way. The whole work starts to have a meaning. Thank you for that.

Response (1): As the reviewer implied, contingencies decreased the amount of data presented here, however, this could have been overcome by the efforts to ensure quality in the analysis of data and interpretation. We thank you for the comment, we are grateful for the positive feedback during our reviewing process that helped to improve our data interpretation, clarity and the overall quality of the manuscript. 

(2) Still, some things regarding the statistics seem to be weird. First, the order of the models in tables 2 and 3 is not logical. Lower AIC values indicate a better-fit model; thus, the order should be changed to show better models at first.

Response (2): The order of the AIC values were re-arranged in Tables 2 and 3 according to the reviewer´s instructions, listed by its best AIC and ED for each distribution (GHG and microorganisms). 

Another problem may be with structural equation modeling (SEM). Structural equation modeling (SEM) aims to define a theoretical causal model consisting of a set of predicted covariances between variables and then test whether it is plausible when compared to the observed data. I cannot see what the basics for the model were; the initial theoretical causal model is missing. In addition, what are the overall parameters for the model to estimate the significance and correctness of the model? What were the p-value, root mean square error of approximation (RMSEA), comparative fit index (CFI), and Tucker–Lewis index (TLI)?

Response (2): Thank you very much for this observation. To clarify this point, in the new version of the manuscript we explained the theoretical causal model for the proposed SEM in the last paragraph of the material and methods section (Lines 306-317). In our work, the SEM model considered latent variables generated with the observed matrix (Env, FG and GHG) and identified the predictivity of the model path. The best environmental variables identified previously by GAM, i.e., conductivity, latitude, nitrate, and pH on GHG and conductivity, OM and nitrite on functional groups (Betaproteobacteria and Methylotenera sp) were used for the Env factor matrix to identify potential influences on the microbial community structure of key functional groups. Then, selected microbial functional groups (FG) influencing GHG concentration were considered for modeling. The path analyses supported our theoretically expected hypothesis with the observed data. 

In this manuscript version the SEM model estimators were included. We initially tested two models, Maximum Likelihood (ML) and diagonally weighted least squares (DWLS). Both models supported an expected path relationship between the latent variables, whereas using DWLS we found out the best model fit performance compared with ML. Therefore the latter was excluded. In the new version of the manuscript, all the parameters evaluating the model performance were included in the results, specifically in the last paragraph (Lines 440-450) and in the Table S7. 

Additionally, we found some minor errors in the manuscript which were highlighted in the new corrected version. 

Line 50- Nitrate and latitude were the best predictors, something was wrong with the previous abstract probably during edit, including independent predictors which are not the best-fit compared with their combined effect. 

Line 68- coma was changed by a period

Line 87 - Authors for the review were included León et al

Table 1 - the title was edited to be more specific.

Line 544 - the functional group matrix was corrected, including Nitrosoarchaeum that was erroneously excluded

LIne 551 - N2O was corrected considering subscript

---

## [Editor Report · Decision Letter 2]

27 Jun 2022

Dissolved greenhouse gases and benthic microbial communities in coastal wetlands of the Chilean coast semiarid region

PONE-D-21-30349R2

Dear Dr. Molina,

We’re pleased to inform you that your manuscript has been judged scientifically suitable for publication and will be formally accepted for publication once it meets all outstanding technical requirements.

Kind regards,

Theodore Raymond Muth

Academic Editor

PLOS ONE
---

## [Editor Report · Acceptance letter]

22 Jul 2022

PONE-D-21-30349R2 

Dissolved greenhouse gases and benthic microbial communities in coastal wetlands of the Chilean coast semiarid region 

Dear Dr. Molina:

I'm pleased to inform you that your manuscript has been deemed suitable for publication in PLOS ONE. Congratulations! Your manuscript is now with our production department. 

Kind regards, 

on behalf of

Dr. Theodore Raymond Muth 

Academic Editor

PLOS ONE